# Location-specific co-benefits of carbon emissions reduction from coal-fired power plants in China

Pu Wang [1,2✉], Cheng-Kuan Lin[3,4], Yi Wang [1,2✉], Dachuan Liu [1,2], Dunjiang Song[1] & Tong Wu[5]

Climate policies that achieve air quality co-benefits can better align developing countries' national interests with global climate mitigation. Since the effects of air pollutants are highly dependent on source locations, spatially nuanced policies are crucial to maximizing the achievement of co-benefits. Using the coal power industry as a case study, this study presents an interdisciplinary approach to assessing facility level co-benefits at every specific source location in China. We find that co-benefits range from US$51-$278 per ton $CO_2$ reduction nationwide and are highly heterogeneous spatially, with "hotspot" regions that should be the priority of emissions reduction policies, and that provinces should use different techno-economic strategies to reduce emissions. The location-specific co-benefit value plus a carbon price serves as a unified environmental indicator that enables policy makers to more accurately understand the social costs of electricity generation from coal burning and provides a scientific framework for geographically nuanced policymaking.

[1] Institutes of Science and Development, Chinese Academy of Sciences, No.15 Zhongguancun Beiyitiao Alley, Haidian District, Beijing 100190, China. [2] School of Public Policy and Management, University of Chinese Academy of Sciences, 19A Yuquan Rd, Shijingshan District, Beijing 100049, China. [3] Department of Environmental Health, Harvard T.H. Chan School of Public Health, 665 Huntington Avenue, Building 1, Room 1406, Boston, MA 02115, USA. [4] School of arts and sciences, Massachusetts College of Pharmacy and Health Sciences, Boston, MA 02115, USA. [5] State Key Laboratory of Urban and Regional Ecology, Research Center for Eco-Environmental Sciences, Chinese Academy of Sciences, Beijing 100085, China. ✉email: wangpu@casisd.cn; wangyi@casisd.cn

Developing countries, such as China and India, face the dual challenge of climate change and air pollution, with the latter usually more urgent and directly associated with domestic welfare[1,2]. Therefore, integrating air pollution control into climate governance can be a highly effective means to motivate developing countries to accelerate climate change mitigation[3]. Reducing $CO_2$ emissions from electric power generation will simultaneously reduce the emissions of sulfur dioxide ($SO_2$), nitrogen oxides ($NO_x$), and primary $PM_{2.5}$. While the effects of carbon emissions are independent of source locations, the effects of air pollutant emissions are highly dependent on source locations, population distribution, climate, and other factors[4,5]. Thus, spatially nuanced policies are key to maximizing the co-benefit of climate policy in air-quality improvement and the related health gains.

The electric power sector is the largest source of carbon emissions in China, accounting for roughly half of the country's total carbon emissions[6,7]. There were over 2000 coal-fired power plants in Mainland China's 31 provinces in 2017, with generation capacity of 980 GW and electricity generation of 4.1 trillion kWh, accounting for 55% of total capacity and 65% of total generation, respectively[8]. Meanwhile, coal-fired plants have been a major cause of the country's serious air pollution in recent years[9,10], contributing to 17%, 19%, and 8% of China's total $SO_2$, $NO_x$, and primary $PM_{2.5}$ emissions, respectively, in 2017[11]. In response, China has adopted three major strategies to reduce the environmental impacts of power plants: eliminating plants with outdated technologies, ultralow emissions retrofitting, and optimizing the siting of plants, and substantial financial and administrative resources have been invested by plant owners and the government[12–14]. When these resources are constrained, all three strategies need to prioritize certain regions, and this prioritization requires location-specific social and economic justifications.

In the past several years researchers have made significant progress in estimating the social costs of air pollution, particularly in terms of the impacts on human health. Most of these studies used atmospheric chemical transport models and exposure–response functions with different assumptions. For example, Anenberg et al.[15] and Lelieveld et al.[2] estimated the global burden of mortality due to air pollution, and found anthropogenic $O_3$ and $PM_{2.5}$ were associated with more than four million premature deaths annually. Other researchers combined the above methodologies with climate mitigation scenarios, and then assessed the co-benefits of climate policies in air pollution reduction and other environmental issues[16–21]. For instance, Shindell et al.[19] investigated methane reduction measures that can reduce global mean warming by 0.5 °C, and estimated that the co-benefit in avoiding premature death was between US$700 to 5000 per ton of methane emissions. West et al.[20] calculated the co-benefits of climate policies under the Representative Concentration Pathway 4.5 (RCP4.5) scenario, and estimated the value to be US$50–380 per ton of $CO_2$ reduction. For China-focused studies, Li et al.[17] applied global models to the latest Chinese data and estimated that a 4% annual reduction in energy intensity would lead to a 24% reduction in $CO_2$ emissions in 2030 relative to the no policy case; such a reduction would require a carbon price of $72 per ton and lead to $340 billion in avoided health costs in 2030; Cai et al.[16] estimated that in 2050 the overall health co-benefits would be 3–9 times of the implementation costs if China fulfills its Paris commitments. One common feature of the existing studies is that they present only aggregated global or national effects of air pollution and the corresponding co-benefit valuations, and lack facility level, location-specific information[22]. Although informative at a macro-level, aggregated results are difficult to translate into implementable policies, which often requires differentiated treatments based on locations and technology types.

To address this gap, we establish an interdisciplinary approach to assess facility-level co-benefits at specific locations. Our model takes advantage of a complete coal plants dataset, the latest fine-scale demographic and health data, and air pollution exposure estimations calibrated specifically for coal plants in China. Our analyses capture technological features, fuel quality, and the specific location of each individual power plant, as well as the relevant demographic, economic, weather, and epidemiological information of affected regions. We find that co-benefits in "hotspot" locations can be up to five times higher than those in more distant locations, and that provinces should use different techno-economic strategies to reduce pollutant emissions. The co-benefit value of per ton $CO_2$ reduction plus a carbon price can serve as a unified indicator that integrates the environmental costs of $CO_2$ and air pollutants, which has straight-forward policy interpretations, and can significantly improve the design of China's emissions trading system, environmental pollution taxes, and other policies aimed at changing the country's fossil-fuel-dominant energy structure.

## Results

**Spatial distribution of co-benefits.** We obtained the locational health co-benefits of per ton carbon emissions reduction in four major steps (see details in "Methods" and Supplementary Fig. 1): (1) using an intake fractions (IFs) model to estimate population exposure to air pollutants from power plants; an IF is defined as the fraction of material or its precursor released from a source that is eventually inhaled by a population; (2) estimating mortality due to increase in exposure to air pollution; (3) economic valuation of air pollution-related mortality based on value of statistical life (VSL) method; (4) calculating co-benefits based on emitting ratios of $CO_2$ and air pollutants. Figure 1 presents the health co-benefits of reducing one ton $CO_2$ emissions from a power plant at each specific location in China. Higher values indicate that phasing out or retrofitting existing plants, or prohibiting the construction of new plants in these areas, will achieve greater co-benefits.

There are four major factors that influence the location-specific health co-benefits: (1) the technological specifications of a power plant, including coal use per unit electricity generation, whether or not it has $SO_2-$, $NO_x-$, and PM-removal facilities, coal quality, etc.; (2) population density and GDP per capita within different ranges from a plant, which affects the value of economic loss caused by air pollutants; (3) baseline mortality rates of diseases related to air pollution, including ischaemic heart disease (IHD), chronic obstructive pulmonary disease (COPD), and lung cancer, which have different relative risk factors corresponding to $PM_{2.5}$ concentration levels; and (4) local climatic conditions, most importantly precipitation levels. For each specific plant, the relative importance of the four factors can vary significantly. We conduct multiple simulations to assess the relative importance of each factor on average (Supplementary Note 1). Overall, population density has the dominant impact on co-benefit values, and precipitation and technology specifications (represented by emission factors) have modest impacts, while baseline mortality rate change has a relatively small impact.

With regional-grid average technology specifications, the building of a power plant at different locations will yield vastly different co-benefit values, ranging from US $51 to 278 per ton $CO_2$ nationwide (Fig. 1). The most prominent "hotspot" region with the highest co-benefit values is around Henan province in central China. The primary reason for this is that air pollutants can be transported across vast distances, affecting populations more than 1000 km away—and Henan is the geographic centre of China's population distribution. Therefore, plants in Henan have maximal influence on air quality of

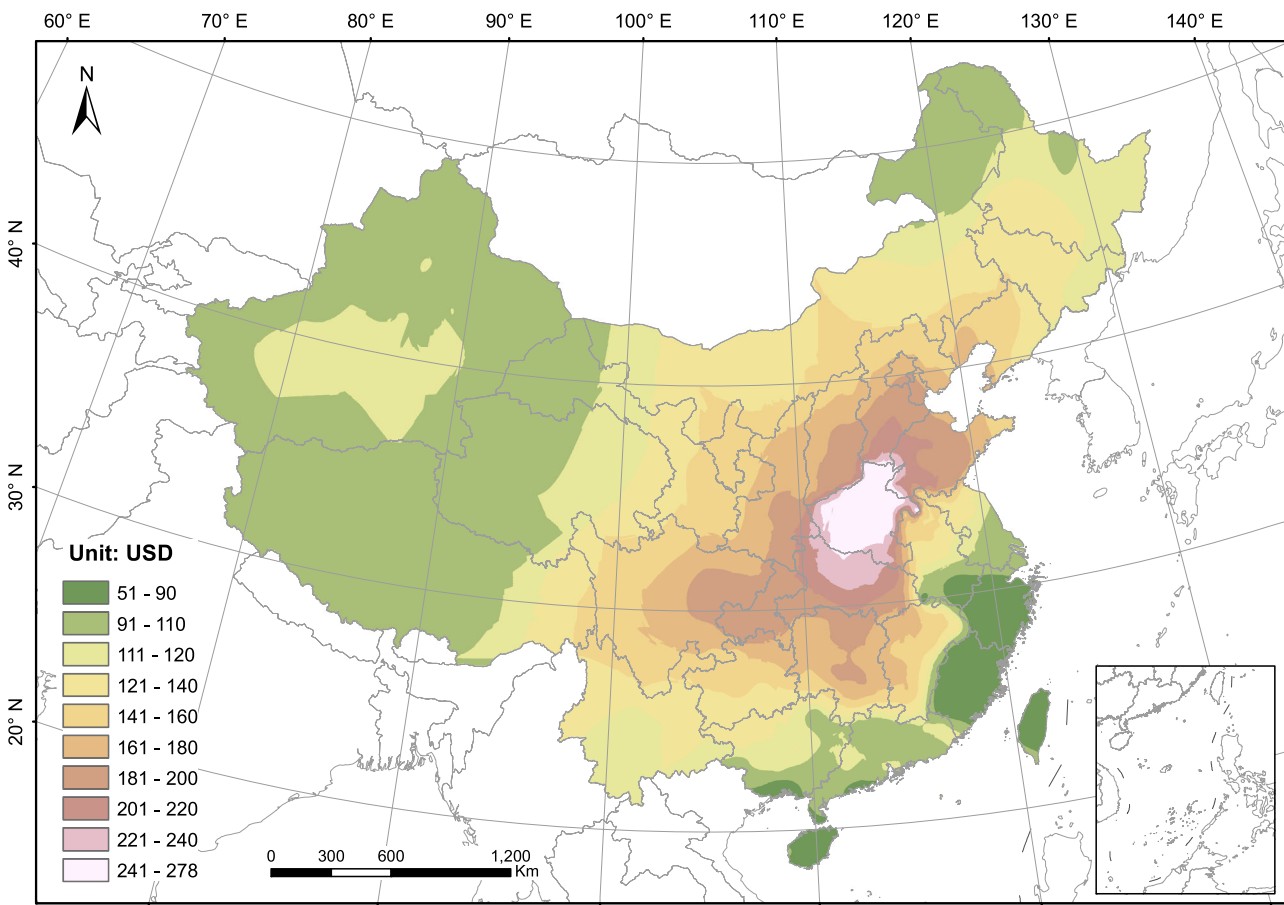

**Fig. 1 Health co-benefits of per ton CO$_2$ emission reduction from coal-fired power plants in different locations.** The unit is in U.S. dollar (USD). Higher values indicate that phasing out or retrofitting existing plants, or prohibiting the construction of new plants in these areas, will achieve greater co-benefits.

the entire country. Besides Henan and adjacent regions, the Beijing–Tianjin–Hebei and Chengdu–Chongqing metropolitan areas also have high co-benefits values, mostly due to their highest population densities in the country. The Shanghai and Hong Kong–Guangzhou–Shenzhen metropolitan areas have the highest population density and GDP per capita in the country, but their values are relatively low, due to a combination of four factors: advanced electricity generators, high quality coal with lower sulfur and ash content, low baseline mortality rates of diseases related to air pollution, and a higher level of precipitation. China's western and northeastern provinces have relatively low values, primarily because of their much lower population densities. The southeastern provinces, such as Zhejiang and Fujian, while populous and economically advanced, have the lowest values, for the same reasons as Shanghai. Overall, Fig. 1 provides a basis for China to set the order of priority for regional energy planning and transitioning away from coal.

We analyse how the pattern in Fig. 1 changes under different assumptions in Supplementary Notes 2 and 3. First, we re-estimate co-benefits using VSLs based on GDP per capita at each city, in contrast to a universal national VSL used in Fig. 1. Second, a power plant can have many detrimental effects on adjacent areas other than PM$_{2.5}$ (e.g., coal ash and heavy metal deposition). We multiply the economic loss within 100 km by three to take into account the other negative local impacts (see "Methods"). Third, we re-estimate co-benefits using different national VSLs and elasticities based on literature review. The corresponding results are presented in Supplementary Figs. 2–6.

**Co-benefits at provincial level.** While results in Fig. 1 enable precise policy making for each specific location, in practice, many

policies are implemented at municipal or provincial scales. For instance, China's carbon emissions trading pilot programmes are operated at the provincial level, and the national emissions trading programme needs to decide allowance allocations across provinces[23]. Thus, we use capacity-weighted mean of co-benefit values for all individual plants in a province to represent the co-benefit value in that province (Fig. 2), and use the same method to calculate the national co-benefit. The national capacity-weighted average co-benefit value is $147. Provincially, again, Henan has the highest co-benefit value at $257, followed by Hubei at $207 and Shandong at $192. Values in Beijing ($180), Tianjin ($183), and Hebei ($183) are not among the highest, even though these provinces had the strictest policies for coal plants phasing-out or retrofitting: using VSLs based on local GDP per capita and higher weights for local effects will yield relatively higher values for these provinces (Supplementary Fig. 4c, d). In general, co-benefits values in the "hotspot" provinces can be 2–4 times higher than those in other provinces, indicating the importance of prioritizing coal use reduction in "hotspot" provinces and using differentiated carbon pricing and environmental taxes at the provincial level.

While Figs. 1 and 2 present the damage caused by power plants located in different regions, Supplementary Fig. 7 presents the economic losses suffered at different locations due to air pollution from all power plants. The annual national loss added up to 460.1 billion USD in 2017 or around 3.7% of the country's total GDP in the same year. Figure 3 illustrates the difference between the rationales behind Figs. 1 and 2 and Supplementary Fig. 7. Figure 3a, c shows the co-benefit of per ton CO$_2$ emission reduction and the damage of annual total emissions, respectively,

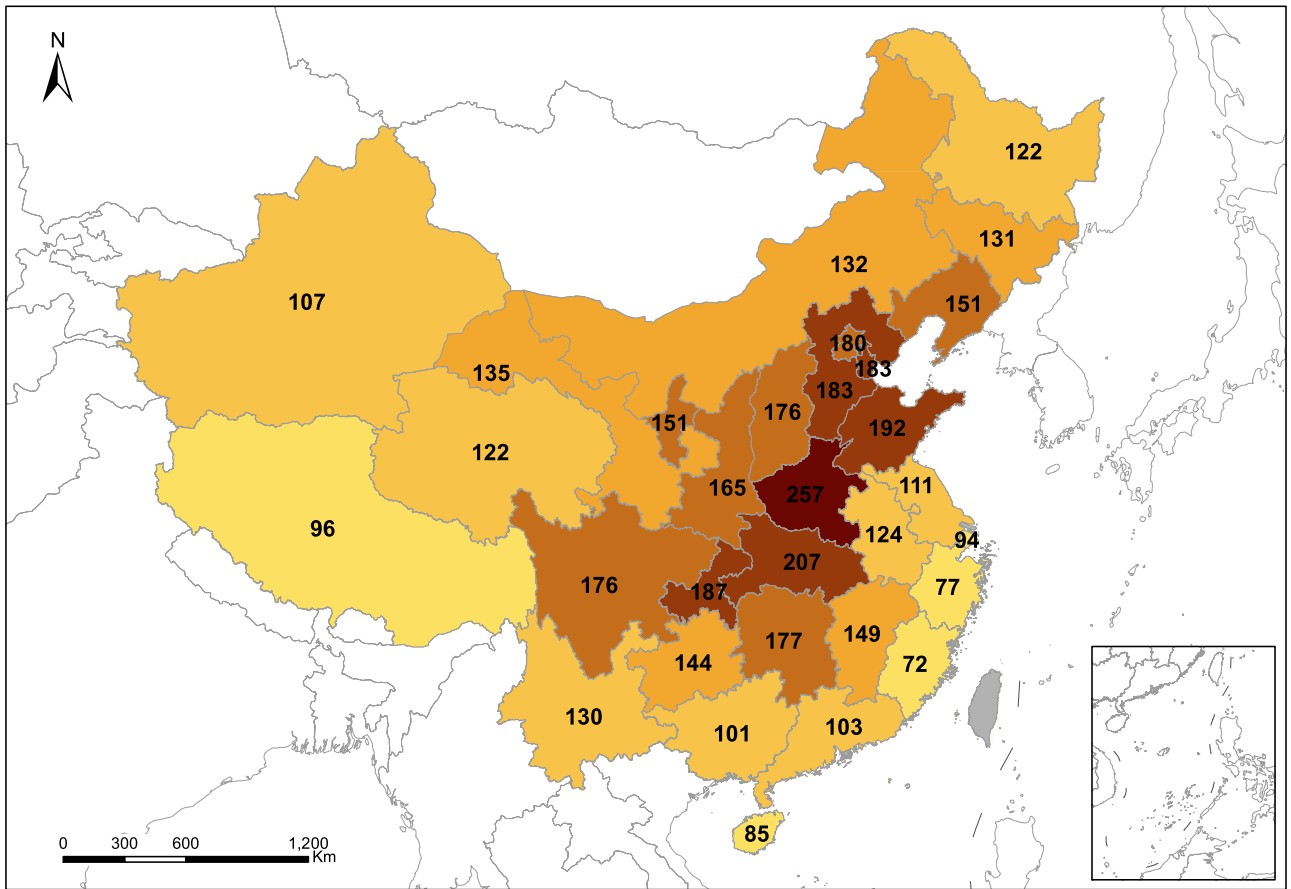

**Fig. 2 Capacity-weighted average health co-benefits of per ton CO₂ emission reduction in different provinces.** The unit is in U.S. dollar (USD). Darker colour indicates that a province has a higher capacity-weighted average health co-benefit value.

from coal-fired plants located in each province; Fig. 3b, d presents the annual per capita loss and the total loss suffered by each province due to coal-fired power generation nationwide. As Fig. 3 shows, the provinces where the most polluting coal plants (inducing the highest losses) are located (Fig. 3a, c) are not necessarily the provinces that suffered the most (Fig. 3b, d). For instance, power plants in Hubei and Chongqing rank the second and the fourth in terms of the average damage, but their per capita losses rank 10th and 14th, respectively. In contrast, power plants in Anhui cause relatively small damages (ranks 20th nationally), but per capita loss in Anhui ranks the 6th. The orders of the total damages and losses in Fig. 3c, d show similar mismatches. The majority of previous studies on health costs of air pollution addressed the latter, namely estimating the losses suffered by affected regions[16,17,20]. However, from the perspective of energy policy targeting, it is more important to address the emission sources that induce the highest costs. Thus, it is arguable that results in Figs. 1 and 2 have more direct policy interpretations in energy planning and siting, as well as in carbon pricing and environmental tax policies.

**Impacts of end-of-pipe technologies on co-benefits.** The health co-benefit of per ton CO₂ emission reduction is calculated according to a power plant's emission factors for CO₂, SO₂, NOₓ, and primary PM₂.₅, measured as grams per kWh electricity generation. These emission factors are determined by coal quality (e.g. ash and sulfur contents of coal), boiler types, and end-of-pipe pollutant removal technologies. In this section we analyse the impacts of four representative technologies that can significantly change these emission factors, including (1) flue gas

desulfurization (FGD) that can reduce up to 95% of SO₂ emissions, (2) selective catalytic reduction (SCR) that can reduce up to 85% of NOₓ emissions, (3) switching from electrostatic precipitators to bag filters (or fabric filters, FAB) that increase the removal efficiency for particulate matters from 93 to 99%, and (4) switching from high-sulfur coal to coal with the national average sulfur content. In this analysis we do not consider any end-of-pipe carbon removal technology, because even though carbon capturing and storage (CCS) technology can substantially change CO₂ emission factors, it is not widely used in coal power sector in China by far. In the future when CO₂ sequestration is widely applied, we would need to take CCS into consideration in our calculations.

At the level of control actually achieved in 2017, the reduction of one kWh coal-fired electricity can reduce 309 grams of standard coal consumption on national average, which simultaneously reduces emissions of 840 g CO₂, 0.43 g SO₂, 0.98 g NOₓ, and 0.15 g primary PM₂.₅. These values vary in each province due to differences in coal quality, energy efficiency of generators, and penetration rates of abatement technologies. Figure 4a presents the air-quality-related health benefit of reducing one kWh coal-fired electricity generation in each province, first based on unabated emission factors for SO₂, NOₓ, and primary PM₂.₅, then based on the ideal operation of FDG, SCR, ESP, and FAB that have removal efficiency of 95%, 85%, 93%, and 99%, respectively, and then based on 2017 actual emission factors. Regional-grid-level CO₂ emission factors in 2017 are given as well. Based on the ratio of carbon and air pollutants emitted per kWh electricity generation, we convert the health benefit values in Fig. 4a to co-benefits per ton CO₂ emission reduction in Fig. 4b.

| a — Health co-benefit per ton CO₂ reduction by each province, unit: USD | | b — Per capita health loss suffered by each province, unit: USD | | c — Total value of health loss caused by plants within each province, unit: Bn USD | | d — Total value of health loss suffered by each province, unit: Bn USD | |
|---|---|---|---|---|---|---|---|
| 1 Henan | 257 | 1 Henan | 494 | 1 Henan | 52.9 | 1 Shandong | 48.3 |
| 2 Hubei | 207 | 2 Shandong | 488 | 2 Shandong | 45.9 | 2 Henan | 48.2 |
| 3 Shandong | 192 | 3 Hebei | 451 | 3 Hebei | 39.1 | 3 Hebei | 33.5 |
| 4 Chongqing | 187 | 4 Shanxi | 422 | 4 Shanxi | 37.1 | 4 Guangdong | 27.0 |
| 5 Tianjin | 183 | 5 Tianjin | 396 | 5 I. Mongolia | 36.2 | 5 Jiangsu | 24.6 |
| 6 Hebei | 183 | 6 Anhui | 389 | 6 Jiangsu | 32.2 | 6 Anhui | 23.9 |
| 7 Beijing | 180 | 7 I. Mongolia | 379 | 7 Guangdong | 24.0 | 7 Sichuan | 23.5 |
| 8 Hunan | 177 | 8 Ningxia | 365 | 8 Anhui | 20.1 | 8 Hubei | 20.3 |
| 9 Sichuan | 176 | 9 Beijing | 344 | 9 Xinjiang | 18.3 | 9 Hunan | 19.7 |
| 10 Shanxi | 176 | 10 Hubei | 343 | 10 Liaoning | 15.0 | 10 Shanxi | 15.6 |
| 11 Shaanxi | 165 | 11 Shaanxi | 337 | 11 Hubei | 14.9 | 11 Liaoning | 15.1 |
| 12 Ningxia | 151 | 12 Liaoning | 333 | 12 Shaanxi | 14.9 | 12 Jiangxi | 14.3 |
| 13 Liaoning | 151 | 13 Gansu | 323 | 13 Guizhou | 14.5 | 13 Guangxi | 13.1 |
| 14 Jiangxi | 149 | 14 Chongqing | 316 | 14 Hunan | 11.9 | 14 Shaanxi | 13.0 |
| 15 Guizhou | 144 | 15 Qinghai | 314 | 15 Zhejiang | 11.0 | 15 Zhejiang | 11.7 |
| 16 Gansu | 135 | 16 Jiangxi | 309 | 16 Jiangxi | 10.7 | 16 Yunnan | 11.1 |
| 17 I. Mongolia | 132 | 17 Guizhou | 308 | 17 Ningxia | 8.9 | 17 Heilongjiang | 11.1 |
| 18 Jilin | 131 | 18 Jiangsu | 302 | 18 Gansu | 8.6 | 18 Guizhou | 11.1 |
| 19 Yunnan | 130 | 19 Hunan | 290 | 19 Jilin | 8.3 | 19 I. Mongolia | 9.7 |
| 20 Anhui | 124 | 20 Xinjiang | 287 | 20 Heilongjiang | 7.8 | 20 Chongqing | 9.4 |
| 21 Heilongjiang | 122 | 21 Jilin | 282 | 21 Tianjin | 7.5 | 21 Fujian | 9.1 |
| 22 Qinghai | 122 | 22 Sichuan | 281 | 22 Fujian | 6.6 | 22 Gansu | 8.6 |
| 23 Jiangsu | 111 | 23 Heilongjiang | 280 | 23 Chongqing | 5.0 | 23 Jilin | 8.0 |
| 24 Xinjiang | 107 | 24 Guangxi | 275 | 24 Shanghai | 4.7 | 24 Beijing | 7.0 |
| 25 Guangdong | 103 | 25 Guangdong | 251 | 25 Sichuan | 3.6 | 25 Xinjiang | 6.5 |
| 26 Guangxi | 101 | 26 Fujian | 240 | 26 Guangxi | 3.4 | 26 Tianjin | 5.3 |
| 27 Tibet | 96 | 27 Yunnan | 233 | 27 Yunnan | 1.8 | 27 Shanghai | 5.2 |
| 28 Shanghai | 94 | 28 Shanghai | 217 | 28 Qinghai | 1.6 | 28 Ningxia | 2.4 |
| 29 Hainan | 85 | 29 Zhejiang | 209 | 29 Hainan | 1.5 | 29 Qinghai | 1.8 |
| 30 Zhejiang | 77 | 30 Hainan | 171 | 30 Beijing | 0.7 | 30 Hainan | 1.5 |
| 31 Fujian | 72 | 31 Tibet | 118 | 31 Tibet | 0.0 | 31 Tibet | 0.5 |

**Fig. 3 The difference between health cost valuations based on emission sources vs. the affected regions. a** Value of health co-benefit from reducing one ton CO₂ emissions from coal-fired power plants located in each province; **b** Per capita health loss suffered by each province due to air pollution caused by all coal plants in China in 2017; **c** total value of health loss caused by coal plants within each province; **d** total value of health loss suffered by each province due to air pollution caused by all coal plants in China in 2017. Provinces connected by blue lines have power plants that are relatively more damaging, but these provinces suffer relatively less; provinces connected by red lines have relatively less damaging power plants, but these provinces suffer more from coal-fired power generation.

Figure 4a, b shows that the co-benefits patterns in Figs. 1 and 2 are not fixed, and should be updated regularly (e.g. annual update) based on changes in penetration rates of different technologies in provinces. From a policy perspective, a province can deploy more end-of-pipe control technologies to change the carbon-air pollutants ratio, thus can change its relative priority in national coal plants phasing out; it can also receive a more favourable policy in a permit trading system or a tax system (see our policy discussion in the "Discussion" section). There is no one-size-fits-all optimal strategy for all the provinces, and detailed local information is needed to improve policy effectiveness at the provincial level. Our calculations in Fig. 4 can provide basis for such updates. First, desulfurization has the biggest impacts on co-benefit values across the country. In particular, in the central and southern provinces where coal has a high percentage of sulfur, desulfurization could reduce up to 80% of pollution costs and substantially change the co-benefit estimates, as in the case of Chongqing. Second, in the northeastern and eastern provinces where coal has lower sulfur contents, desulfurization would only reduce roughly 35–55% of pollution costs. Consequently, in these areas the application of NO$_x$ and dust removal technologies become important. Third, in provinces where pollution costs are high, such as in Chongqing and Guizhou, switching to coal with better quality can also have a significant impact on co-benefit values. Overall, we find the national emissions reduction due to

end-of-pipe control technologies was around 70% on average in 2017, which is consistent with previous studies[11].

**Analysis of pollutant-specific co-benefits.** The emissions from a power plant can have health effects in regions both nearby and far away. Figure 5 presents the spatial distribution of pollutant-specific co-benefits from plants located in six regional grids. The affected regions are divided into four ranges: within 100 km, 100–500 km, 500–1000 km, and more than 1000 km. The costs of primary PM$_{2.5}$ represent only around 15% of total costs in all ranges, much smaller than the proportions of SO$_2$ and NO$_x$. In most grid regions, the majority of costs from NO$_x$ (roughly between 50 and 55%) occurred in the 100 km (local) and 100–500 km (mostly within a province) rings, while the majority of costs from SO$_2$ (between 65 and 90%) occurred in the 500 km and 1000plus km rings, indicating that more benefits from de-NO$_x$ technologies are captured locally and within a province, while the benefits of desulfurization have greater "spill-over" effects to other provinces. Comparing the six grid regions, relatively large portions of the co-benefits in the north, east, and central grids occurred locally and within-province, while in the other grids, particularly in the northwest, the co-benefits are mostly captured in the 500–1000 km and 1000plus km rings. This indicates that in less populated regions (also the less developed regions) such as the northwest, while the costs of pollutant

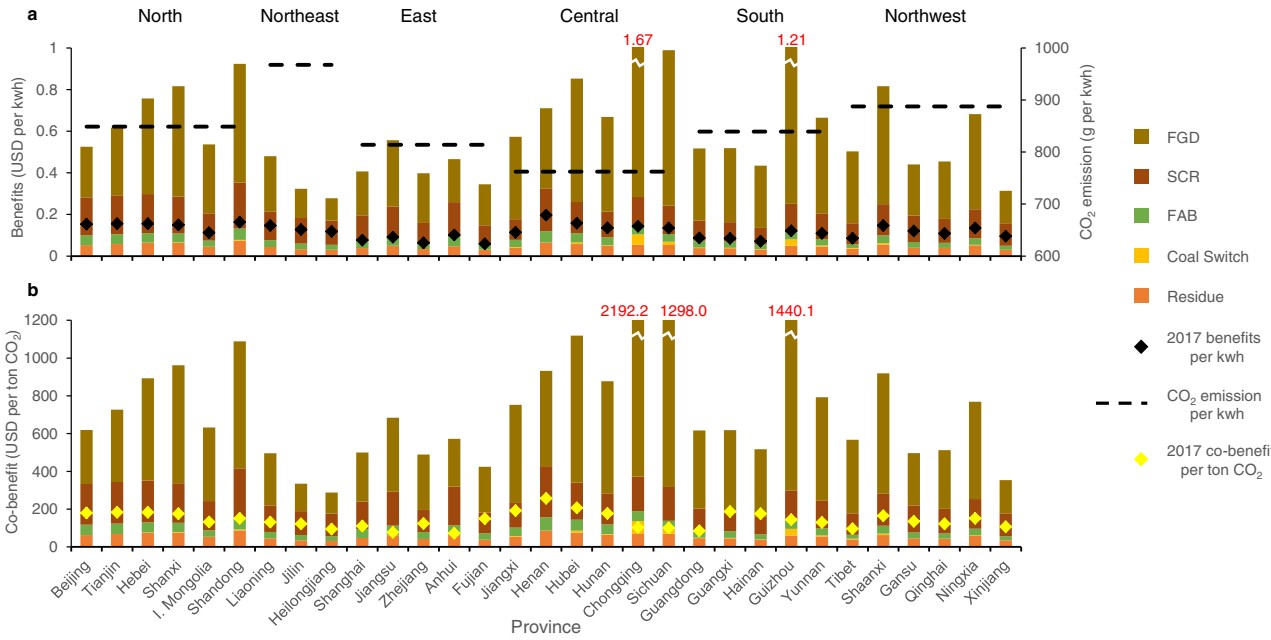

**Fig. 4 Potential benefits of emission reduction technologies and the impacts on co-benefit values. a** Potential benefits of four emission reduction methods in reducing health costs per kWh electricity generation. **b** Impacts of four emission reduction methods on co-benefit values per ton $CO_2$ emission. The four representative methods to reduce emissions are (1) flue gas desulfurization (FGD); (2) selective catalytic reduction (SCR); (3) switching from electrostatic precipitators to fabric filters (FAB); and (4) switching from high-sulfur coal to coal with the national average sulfur content. "Residue" represents the value under the ideal situation when all methods are perfectly adopted. The black and yellow diamond-shaped dots represent the actual provincial values in 2017.

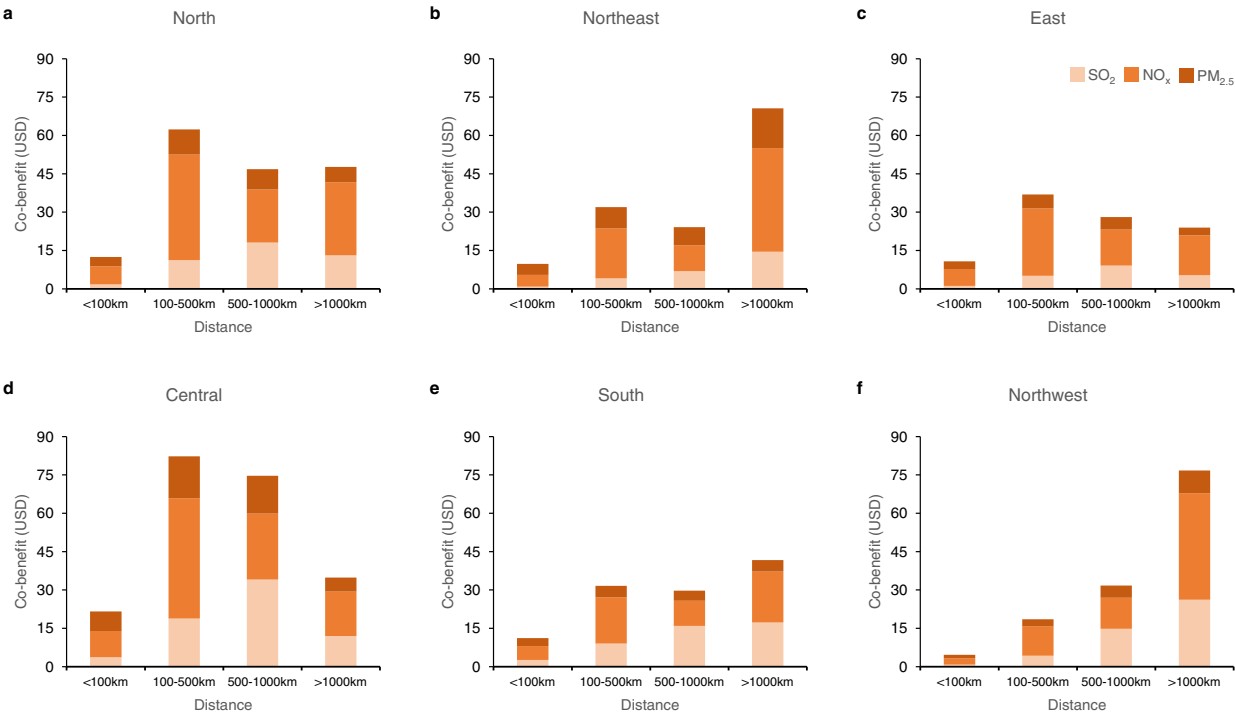

**Fig. 5 The distribution of air pollutant-specific co-benefits in different ranges.** The affected regions are divided into four ranges according to distance from power plants: within 100, 100–500, 500–1000, and more than 1000 km. **a–f** present the results in six regional power grids, respectively (unit is in USD per ton $CO_2$ emission reduction).

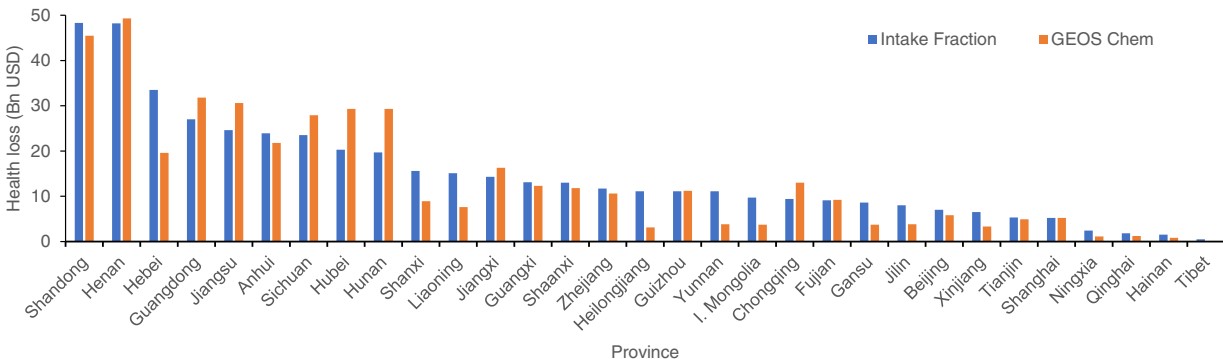

**Fig. 6 Values of provincial health losses in 2017 based on intake fractions (IF) model and GEOS-Chem model.** IF model results have higher values in remote regions, such as Heilongjiang, Inner Mongolia, Yunnan, and Tibet, while the GEOS-Chem model results have higher values in central provinces, such as Hubei, Hunan, and Sichuan.

removal are usually borne locally, the adoption of such technologies largely benefit other provinces. Therefore, besides a strict application of the polluter-pays-principle by imposing a pollution tax equal to damage, it is necessary to design compensation and cost-sharing mechanisms to encourage the adoption of pollution reduction technologies in less developed regions.

**Comparison of results based on different models.** IF model is time-efficient to process more than 2000 simulations for all coal-fired plants in China, and the model parameters are calibrated specifically for China's power plants, though it only considers variables that have the strongest predictive power. We use the GEOS-Chem model to statistically test the accuracy of the IF model. GEOS-Chem is a three-dimensional air-quality model that takes into account fine-scale emissions, atmospheric chemistry, aerosol microphysics, precipitation, wind speed and direction, and other input data. Identical national power plants inventories are used in IF and GEOS-Chem simulations. The simulation results show that national total health losses due to coal-fired plants in 2017 was 460.1 billion USD based on the IF model, and was 426.4 billion USD based on GEOS-Chem (Supplementary Figs. 7 and 8). We then calculate health losses suffered by each province in China based on the two models (Fig. 6). IF model results have higher values in remote regions, such as Heilongjiang, Inner Mongolia, Yunnan, and Tibet, while the GEOS-Chem model results have higher values in central provinces, such as Hubei, Hunan, and Sichuan. In majority of the other provinces, the values from the two models are close. We also compare GEOS-Chem simulation results with the actual observation data from 1444 air-quality monitoring stations across China (Supplementary Fig. 9). Supplementary Figure 9 indicates that GEOS-Chem tends to underestimate $PM_{2.5}$ concentrations in remote provinces, while overestimate $PM_{2.5}$ concentrations in central provinces. Thus, the large differences between the two models' results in the seven provinces mentioned above can be partly explained by GEOS-Chem's biases. Next, we calculate health losses suffered by each of the 2761 Chinese county-level districts according to the two models, and run linear regressions for counties within each of the six regional power grids (Fig. 7). All the six regressions show good correlation between the results based on the two models ($R^2$ range from 0.73 to 0.91). Thus, overall, we find that the IF model has the advantage of time efficiency in processing a large number of simulations and its results are reliable for policy analysis at large geographic scales. If the government wants to calculate an externality tax for an individual plant based on more comprehensive variables, then the IF model is not sufficiently customized, and a more sophisticated model, such as GEOS-Chem, should be used.

**Discussion**

Targeting the co-benefits of carbon emissions reduction in air pollution control has been an important principle for China's climate policy making. China's key strategies to reduce carbon and air pollutants emissions from the power sector, such as phasing out power plants in key regions and/or with outdated technologies, the adoption of end-of-pipe control facilities, optimizing the siting of plants, and various market-based policy instruments, will all benefit from location-based calculation of co-benefits.

This paper makes two contributions to policymaking that aims to integrate location-based information. First, our results provide a technically feasible approach to quantifying the total benefits of reducing various pollutant emissions from a power plant in any location. Coal-fired power plants are among the largest sources of $CO_2$ and air pollutants worldwide, but without policy interventions, they remain the most affordable and accessible electricity providers in many developing countries. The co-benefit value of per ton $CO_2$ reduction plus a carbon price can serve as a unified environmental indicator that enable policy makers to more accurately understand the social costs of electricity generation from coal burning, and to better address regional energy planning and environmental policymaking related to power plants. International Energy Agency (IEA)'s study estimated that in 2020, without a carbon price, the levelized cost of electricity (LCOE) for coal power in China was 5.2 US cents per kWh, while the LCOE for wind and solar power were 5.8 and 5.1 US cents per kWh, respectively[24]. Thus, adding a carbon price and air pollution costs to coal power price is important to help enhance the relative market competitiveness of wind and solar power and reduce coal power. Moreover, China's environmental policies in the past have featured one-size-fits-all designs[25], such as mandating the same retrofitting technology for coal-fired plants in all provinces. But such policies have imposed heavy and unsustainable costs to the government, plants owners, and end users. Our study provides quantitative basis to support the design of geographically nuanced policies to phase out or retrofit coal-fired plants in a cost-effective manner, and develops a scientifically sound framework in which provinces can choose differentiated mitigation strategies that best fit their local conditions.

Second, our study can help improve the design of China's carbon pricing and pollution tax policies. In recent years China has been promoting market-based policies in climate and

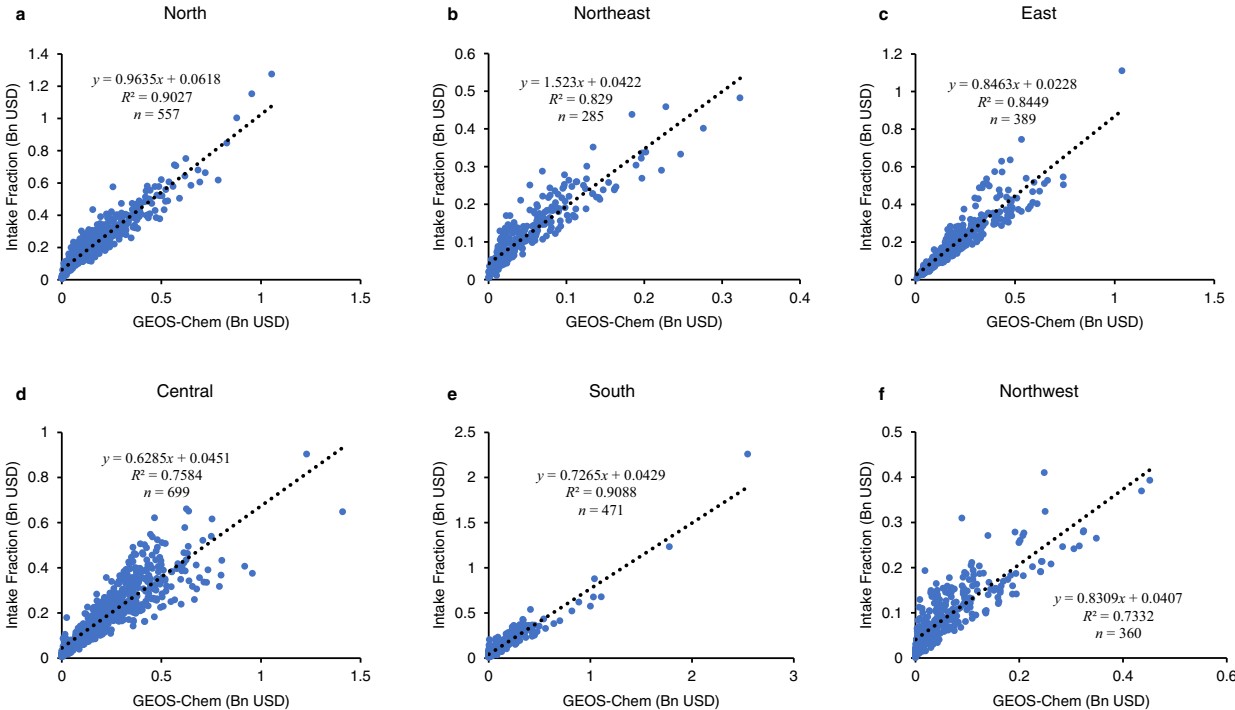

**Fig. 7 Linear regressions between results based on intake fractions model and GEOS-Chem model.** We calculate health losses suffered by each of the 2761 Chinese county-level districts according to the two models, and run linear regressions for counties within each of the six regional power grids. **a–f** present the results in six regional power grids, respectively.

environmental fields[26], including national and provincial carbon emissions trading systems and environmental pollution taxes. According to previous studies, the global social cost of one ton $CO_2$ emission is around US\$20–\$50[27]. The social costs of air pollution are complicated by the heterogeneous influences of source locations and socioeconomic surroundings. The results of our study provide locational health co-benefits and suggest that it is possible to integrate carbon pricing and pollution taxes into one unified pricing scheme, which could be either an emissions trading system or a tax. Muller and Mendelsohn[5] argue that setting location-specific trading ratios in $SO_2$ and $NO_x$ emissions trading systems can achieve significant economic benefits. In parallel, our study suggests that similar trading ratios in a carbon emissions trading system can enhance the achievements in health co-benefits. For instance, to compensate for one ton $CO_2$ emission from a plant in Henan, the plant owner will need to buy allowances for two tons of $CO_2$ emissions from Ningxia, or three tons from Qinghai, since the co-benefit value is much higher in Henan. Similarly, under a unified emissions tax policy, one ton carbon emission in Henan would be taxed twice as much as one ton in Ningxia, or three times as much in Qinghai. Since China's Ministry of Ecology and Environment is in charge of measuring and pricing both carbon and air pollutants emissions, and the carbon and air pollutants charges are imposed on the same power plants, a unified pricing scheme can streamline the administrative processes, and send market signals that can more comprehensively reflect the social costs of coal power.

A series of challenges need to be addressed in order to better integrate our analysis into actual policies. First, the current models still have large uncertainties in assessing exposure and health losses, which can undermine the benefits of location-specific policies[4]. Even state-of-the-art models such as GEOS-Chem also show significant discrepancies between predicted and observed data. Therefore, our results demonstrate the possibility

of precise policy making using plant-specific carbon price and co-benefits, though with relatively large uncertainties; in the future when better modelling techniques and real-time monitoring data are available, such a policy will become realistic. Currently, our results can be more realistically used in larger scale policy making, such as in national energy planning, including the approval, retrofitting, and shutdown of coal-fired power plants in different locations; and in provincially differentiated carbon pricing policies, such as emissions trading systems or carbon tax policies. Second, we recognize that this study, as well as previous studies on co-benefits, only take into account mortality caused by increase in $PM_{2.5}$ concentration, and does not consider other types of damages, such as higher morbidity and lower work productivity. This problem can be addressed by adding more comprehensive valuation analyses in future models. Third, a series of practical questions need to be addressed during policy implementation, including how often the co-benefits valuation should be updated based on climate, demographic, and epidemiological changes, how to treat plants close to provincial boundaries that are charged differently, and how to prevent rent-seeking in the process of assigning different prices to different plants. This type of issue is not unique for the policies proposed in this study, and can be addressed by administrative discretion and referring to precedents. And fourth, policy makers need to make various political and ethical decisions about valuation parameter choices and fairness concerns. Choosing different benchmark VSLs or elasticities will significantly change the values of co-benefits, thus affecting the stringency of coal restriction polices (Supplementary Fig. 4); adjusting VSLs based on income levels in different regions can generate more precise locational estimations, but faces significant ethical challenges (Supplementary Figs. 2 and 3); socio-political considerations can outweigh climate and environmental concerns; for instance, as the national capital city, Beijing has the strictest coal control policies, even

though it does not have the highest estimated co-benefits. Such social and political decisions are beyond the scope of our analysis, though our calculation based on environmental benefits can serve as a benchmark for actual policy making.

Overall, the methods developed in this study represent a scientifically sound approach to connecting facility-level information to macro-level policy-making by means of data-intensive analysis. Going forward, it is possible to modify these methods and apply them to similar issues, such as assessing the locational impacts of power systems on water resources and land use, thereby capacitating more cost-effective environmental policies with spatial nuances in mind.

## Methods

**Datasets.** The information on geographic coordinates, technology type, and installed capacity of each individual coal-fired power plant in China was extracted from the open-source dataset, "Existing coal plants in China" compiled by Global Energy Monitor's Global Coal Plant Tracker[28], with each plant's information verifiable through online satellite maps and power companies' official websites. We also used the Carbon Monitoring for Action (CARMA) dataset[29] developed by the Center for Global Development to complement the Global Energy Monitor dataset. Our final dataset included 2475 power plants (for some plants each generation unit is a single entry), with total installed capacity of 914 GW, which represents 93% of mainland China's coal fleet in 2017 (Supplementary Fig. 10).

Population distribution data are based on the 1×1 km global population grid data for 2017 developed by Oak Ridge National Laboratory's LandScan project, which is the finest-grain population distribution data to date. China's national and prefecture level GDP data were extracted from Chinese statistical yearbooks for the year 2017. Province-level mortality rates for diseases associated with ambient air pollution were based on Zhou et al.'s study[30], which was part of the Global Burden of Disease (GBD) project (Supplementary Table 1). Precipitation data between 2011 and 2016 are based on hourly meteorological monitoring data from 839 stations across China[31] (Supplementary Fig. 11).

**IFs of air pollutants.** An IF is defined as the fraction of material or its precursor released from a source that is eventually inhaled or ingested by a population[32], which provides a straight-forward summary measure of the impacts of a polluting source, such as a power plant. Zhou et al. estimated China-specific coefficients for IF as the dependent variable, and distance from coal-fired power plants, population density, and climate as independent variables, using 29 coal plants data across the country[33] (Supplementary Table 2).

Following Zhou et al.'s study and using a similar method as Parry et al.[3], we calculated IFs of primary and secondary $PM_{2.5}$ formed from emissions of $SO_2$, $NO_x$, and dust from coal plants. For each power plant, the affected regions are divided into four ranges according to the distance from the power plant: within 100, 100–500, 500–1000, and beyond 1000 km (Supplementary Fig. 10). Then, Geographic Information System (GIS) modelling was used to calculate the population covered by each distance range based on the LandScan 1 × 1 km population distribution data. By definition, IF can be expressed in Eq. (1):

$$\mathrm{IF}_{i,j,r} = \frac{\sum_{g=1}^{N} \mathrm{Pop}_g \times \triangle C_{i,j,r} \times \mathrm{BR}}{Q_{i,j}} \quad (1)$$

where the subscripts $i$, $j$, $g$, $r$ represent power plants, pollutant types (i.e., $SO_2$, $NO_x$, or primary $PM_{2.5}$), 1 × 1 km population grids, and ranges from a power plant, respectively. $Q_{i,j}$ is the emission rate of pollutant $j$, or in the case of secondary $PM_{2.5}$, its precursor $j$, from power plant $i$; $\mathrm{IF}_{i,j,r}$ is the IF of pollutant $j$ from power plant $i$ in range $r$; $\mathrm{Pop}_g$ is the population in grid $g$; $\triangle C_{i,j,r}$ is the change in ambient concentration of $PM_{2.5}$ ($\mu$g per m$^3$) in range $r$ caused by emissions of pollutant $j$ from power plant $i$. BR is the population-average breathing rate (m$^3$ per day), and is assumed to be 20 m$^3$ per day, following Zhou et al.[33].

While many specific climate factors, such as temperature, wind speed and direction, and relative humidity, can influence the dispersion, deposition, and chemical reactions of pollutants, previous studies have found them insignificant in affecting IF over a large area[34]. Zhou et al.[33] identified precipitation as the most significant climate factor that has a negative correlation with IFs and can explain most of the residuals correlated with climate factors. We calculated average annual precipitation between 2011 and 2016 based on daily meteorological monitoring data from 839 stations across China, and used Kriging interpolation to estimate annual precipitation at each location in China (Supplementary Fig. 11). Using coefficients in Supplementary Table 2, we calculated $\mathrm{IF}_{i,j,r}$ caused by $SO_2$, $NO_x$, and primary $PM_{2.5}$ emissions based on Eq. (1).

**Estimating the relative risks of studied diseases.** Following Anenberg et al.[15], West et al.[20], and Burnett et al.[35], the exposure–response relationship in this study focus on increased mortality rates of three types of diseases, namely IHD, COPD, and lung cancer, due to increase in annual average $PM_{2.5}$ concentration. For a 10 $\mu$g per m$^3$

increase in annual average $PM_{2.5}$ concentration, the relative risks for both IHD and COPD are 1.13 (95% confidence interval [CI]: 1.10–1.16), and the relative risk for lung cancer is 1.14 (95% CI: 1.06–1.23).

In Eqs. (2) and (3), $RR_{k10}$ is the relative risk of disease $k$ in correspondence to a 10 $\mu$g per m$^3$ increase in annual average $PM_{2.5}$ concentration. For people living in grid $g$, $RR_{g,k,i,j}$ is the relative risk of disease $k$ in correspondence to the change in ambient $PM_{2.5}$ concentration ($\triangle C_{i,j,r}$) caused by pollutant $j$ emitted from power plant $i$, while grid $g$ is located in range $r$ from plant $i$; $RR_{g,k,r}$ is the relative risk of disease $k$ in correspondence to the change in ambient $PM_{2.5}$ concentration caused by all power plants within range $r$ from grid $g$.

$$\mathrm{RR}_{g,k,i,j} = \mathrm{RR}_{k10}^{\left(\frac{\triangle C_{i,j,r}}{10}\right)} \quad (2)$$

$$\mathrm{RR}_{g,k,r} = \mathrm{RR}_{k10}^{\left(\frac{\sum_{i=1}^{n} \sum_{j=1}^{3} \triangle C_{i,j,r}}{10}\right)} \quad (3)$$

**Impacts on mortality rates.** Mortality rates attributable to increased $PM_{2.5}$ concentration caused by every single coal-fired power plant can be estimated by applying GBD methods[36]. In the GBD method, population attributable fraction (PAF) is defined as the percentage of disease mortality attributable to certain exposure in a population. PAF is estimated as

$$\mathrm{PAF}_{g,k,i,j} = \frac{P_{g,k,i,j} \times \mathrm{RR}_{g,k,i,j} - P'_{g,k,i,j} \times \mathrm{RR}'_{g,k,i,j}}{P_{g,k,i,j} \times \mathrm{RR}_{g,k,i,j}} \quad (4)$$

$$\mathrm{PAF}_{g,k,r} = \frac{P_{g,k,r} \times \mathrm{RR}_{g,k,r} - P'_{g,k,r} \times \mathrm{RR}'_{g,k,r}}{P_{g,k,r} \times \mathrm{RR}_{g,k,r}} \quad (5)$$

where in Eq. (4), $\mathrm{PAF}_{g,k,i,j}$ represents the fraction of a specific disease ($k$) mortality in grid $g$ attributable to pollutant $j$ emitted from power plant $i$; $P_{g,k,i,j}$ is the proportion of population exposed to air pollutant $j$ emitted from plant $i$, presumably equal to one (=100%) due to the universal impacts of air pollution; $P'_{g,k,i,j}$ is the counterfactual proportion of the population not exposed to emissions from plant $i$, which is equal to one as well. $\mathrm{RR}'_{g,k,i,j}$ is the counterfactual risk from not being exposed to plant $i$, equal to one. Thus, the equation can be simplified as

$$\mathrm{PAF}_{g,k,i,j} = \frac{\mathrm{RR}_{g,k,i,j} - 1}{\mathrm{RR}_{g,k,i,j}} \quad (6)$$

$$\mathrm{PAF}_{g,k,r} = \frac{\mathrm{RR}_{g,k,r} - 1}{\mathrm{RR}_{g,k,r}} \quad (7)$$

Similar denotation rules apply to Eqs. (5) and (7). Disease-specific mortality change can be estimated as

$$\triangle \mathrm{Mortality}_{g,k,i,j} = \mathrm{mortality}_{g,k} \times \mathrm{PAF}_{g,k,i,j} \quad (8)$$

$$\triangle \mathrm{Mortality}_{g,k,r} = \mathrm{mortality}_{g,k} \times \mathrm{PAF}_{g,k,r} \quad (9)$$

where $\mathrm{mortality}_{g,k}$ is the age- and sex-standardized mortality rate (2012 China national standardized population) of a specific disease $k$ in grid $g$; $\mathrm{PAF}_{g,k,i,j}$ is the fraction of mortality attributable to pollutant $j$ emitted from plant $i$. $\triangle \mathrm{Mortality}_{g,k,i,j}$ is the fraction of mortality attributable to emissions of pollutant $j$ from plant $i$. The distributions of age and sex in grid $g$ are assumed to be homogeneous across all regions in the same province. Similar denotation rules apply to Eq. (9).

**Valuation of air pollution-related mortality.** The VSL method, which is based on surveys on people's willingness to pay to avoid fatality risks[37], was used to estimate the economic loss of increased mortality caused by air pollution. We used the VSL suggested by the US EPA[38] as the baseline ($VSL_{base}$), which is $7.4 million per person in 2006 value, and adjusted it to 2017 value. VSL in each grid $g$ was calculated based on China's national and prefecture level socioeconomic data according to Eq. (10):

$$\mathrm{VSL}_g = \mathrm{VSL}_{base} \times \left( \mathrm{pcGDP2017}_g / \mathrm{pcGDP2017}_{US} \right)^{0.5} \quad (10)$$

Here, $\mathrm{pcGDP2017}_g$ is GDP per capita of grid g in 2017, calculated from China's national and prefecture level GDP and population data in 2017; $\mathrm{pcGDP2017}_{US}$ is GDP per capita in the United States in 2006, adjusted to 2017 value taking into account of inflation. Income elasticity of the VSL with respect to per capita GDP is 0.5, following West et al.[20]. The exchange rate in 2017 was one dollar for 6.75 Chinese Yuan (CNY).

Using the results in the increase in mortality rates, we calculated the economic loss caused by air pollutants from each power plant $i$ and in each grid $g$. For plant $i$, emissions of pollutant $j$ will cause the mortality rate in grid $g$ to increase by $\triangle \mathrm{Mortality}_{g,k,i,j}$. The air pollution-associated economic loss caused by plant $i$ can be calculated by the following equation:

$$L_i = \sum_{k=1}^{3} \sum_{r=1}^{4} \sum_{j=1}^{3} \sum_{g=1}^{N} \mathrm{VSL}_g \times \mathrm{Pop}_g \times \triangle \mathrm{Mortality}_{g,k,i,j} \quad (11)$$

where $L_i$ is the nationwide economic loss attributable to plant $i$; for residents of grid $g$, the air pollution-associated economic loss caused by all coal-fired power plants in China is estimated as

$$L_g = \sum_{k=1}^{3} \sum_{r=1}^{4} VSL_g \times Pop_g \times \triangle Mortality_{g,k,r} \qquad (12)$$

Equations (11) and (12) are also used to calculate the economic loss from different combinations of pollutant types and geographic ranges, which can yield more nuanced results for policy targeting (Figs. 3–5).

As comparisons to the main results using a universal VSL based on national per capita GDP and an income elasticity of 0.5, we also calculated co-benefits using VSLs based on alternative assumptions, including (1) an income elasticity of 1; (2) prefecture-level VSLs based on local income; and (3) VSL based on a contingent valuation survey study from China[39]. These results are presented in Supplementary Notes 2 and 3.

On national average, the distribution of economic loss caused by a plant across the four ranges (within 100, 100–500, 500–1000, and beyond 1000 km) is roughly 1:3:3:3. Since a power plant can have many detrimental effects on adjacent areas other than $PM_{2.5}$ (e.g., coal ash and heavy metal deposition), the economic loss within 100 km could be underestimated. As a comparison with the original results, in Supplementary Fig. 3 we multiply the economic loss within 100 km by 3 to make it roughly the same as the losses in other ranges. The choice of weight for local impacts can be adjusted based on various environmental and socioeconomic considerations, and from a regional planning perspective, policy makers can give higher weights to local impacts in order to avoid construction of coal plants in metropolitan areas.

We use the GEOS-Chem model (GCHP 13.0.2 version) to statistically test the accuracy of the IFs model. We created a national inventory for pollutants emissions from all coal-fired power plants in China. Inventories for other economic sectors are from the Multi-resolution Emission Inventory for China (MEIC)), a bottom-up emission inventory framework developed and maintained by Tsinghua University (http://meicmodel.org/). The model was run with a horizontal resolution of approximately 0.5° × 0.5° in China (internal cubed sphere resolution) and 72 vertical layers. We then compared GEOS-Chem results with IF model results (Figs. 6 and 7) as well as with observational data for annual average $PM_{2.5}$ concentration from 1444 air-quality monitoring stations across China (Supplementary Fig. 9).

**Distribution of co-benefits across geographic regions**. When technology settings and coal quality are given for a plant, the co-benefit of per ton $CO_2$ emissions reduction were calculated according to a power plant's emission factors for $CO_2$, $SO_2$, $NO_x$, and primary $PM_{2.5}$. China's electricity grids are divided into six major regional grids (Supplementary Fig. 12). Several previous studies have developed complete emissions inventories for coal-fired power plants in China, such as the MEIC. Liu et al.[40] calculated regional-grid level average emission factors for $CO_2$, $SO_2$, $NO_x$, and $PM_{2.5}$ per kWh of electricity generation in 2010 based on MEIC (Supplementary Table 3). From 2010 to 2017, national average per kWh coal use decreased from 333 to 309 g, while emission factors of $SO_2$, $NO_x$, and $PM_{2.5}$ decreased by 83%, 63%, and 44%, respectively[11]. The 2017/2010 ratios are used to estimate regional-grid level emission factors in 2017 (Supplementary Table 4). Since our study aims to estimate co-benefits of a power plant at a specific location with average technology settings and fuel quality in the region, the use of regional grid average emission factors is appropriate. Unabated emission factors for $SO_2$ in each province were calculated based on per kWh coal consumption and sulfur contents of coal. The sulfur content data and unabated emission factors for $NO_x$ and primary $PM_{2.5}$ are from Liu et al.'s study[40]. The removal efficiency of four representative technologies that can reduce pollutants emissions, including FDG, SCR, ESP, and FAB, are assumed to be 95%, 85%, 93%, and 99%, respectively, based on existing literature[41–43].

After we obtained co-benefit values for 2475 plants/generation units, a Kriging interpolation algorithm was applied to estimate the geographic distribution of co-benefits from a power plant built with regional grid average technology settings and fuel quality. The provincial average co-benefit values were obtained by taking a capacity-weighted mean of co-benefit values for all individual plants in that province (Fig. 2). Since China's coal power industry adopts the "equal shares dispatch" rule, meaning that all power plants in a province are assigned roughly the same operating hours each year[44], the weights can also approximate each plant's share in total electricity generation. Similar procedures were also applied to the calculation of regional grid average co-benefits (Fig. 5).

**Reporting summary**. Further information on research design is available in the Nature Research Reporting Summary linked to this article.

## Data availability

Source data are provided with this paper. Other data that support the findings of this study are available from the corresponding author upon reasonable request.

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

## Acknowledgements

This study is supported by the Project No. 72004216 (to P.W.) funded by National Natural Science Foundation of China (NSFC).

## Author contributions

P.W. conceived of and designed the study. P.W., C.-K.L., D.L. and D.S. analysed data. P.W. led the writing of the paper and Y.W. and T.W. contributed to the review and revision.

## Competing interests

The authors declare no competing interests.
