## [Peer Review File · Nature Communications]

Peer Review comments, initial round review –

Reviewer #1 (Remarks to the Author):

This is a good paper analyzing Location-specific co-benefits of carbon emissions reduction from coal-fired power plants in China and therein some economic and non-economic concerns. However, I recommend following changes and/or explanations.

1. The first section heading of Introduction needed to be placed below Abstract.
2. In the first section provide worth citing references at the end of the sentence with the given addition of fossil fuels share in energy mix

While the effects of carbon emissions are independent of source locations, the effects of air pollutant emissions are highly dependent on source locations, population distribution, fossil fuels shares in energy mix, climate, and other factors (Shakeel and Zeshan, 2020; Shakeel, 2020).

3. Though you provided the review in your introduction yet there is need to explain your analytical framework/empirical models with the help of equations and explanation. If you find that you have provided enough information in the methodology then it may be skipped.

4. The section of data and variable explanation will also be needed as a separate or sub section of the empirical models/methodology. Please note that in a research article, there are at least five sections namely; Introduction, Review of Literature, Empirical/Theoretical Model and Data description, Results and conclusion. Do provide these in your paper.

5. You provided methodology after the results while it should be before or after the theoretical model and data description section as elaborated before.

6. Do provide the conclusion of the study as indicated in point no. 4 above.

7. Similarly check for spelling/grammar mistakes and also explain why CO₂ equivalence were not used because there are also other gases like chloropholoro carbons, methane gas etcetera and thus CO₂e would be a better estimate of pollution instead of CO₂? You can use footnote for this explanation.

References

Zeshan, M. Shakeel, M. (2020) Adaptations and Mitigation Policies to Climate Change: A Dynamic CGE-WE Model. Singapore Economic Review.

DOI: <https://doi.org/10.1142/S0217590820500654>

Shakeel, M. (2020) Economic output, export, fossil fuels, non-fossil fuels and energy conservation: evidence from structural break models with VECMs in South Asia. Environmental Science and Pollution Research. <https://doi.org/10.1007/s11356-020-10729-9>

Reviewer #2 (Remarks to the Author):

General comments

The article investigates the location-specific health co-benefits of policy measures implemented at

the provincial level to reduce carbon emissions from coal-fired power plants in China. Health co-benefits are given in USD per-ton CO₂ emissions reduction and represented in maps and therefore form a sound information basis for political decision makers (compared to otherwise abstract physical metrics).

By means of the applied methodology it is possible to identify areas (and even individual plants, however, with regional-grid-average technology specifications) with a high potential for generating co-benefits resulting from investments in carbon mitigation. A strength of the article is the interdisciplinary approach, the combination of economic, ecological and social indicators to evaluate the issue. The authors integrated state-of-the-art concepts for their holistic investigation. Selected input data are actual and representative for the case study.

Concerning the results, the shown interrelationship between air pollutant-specific costs and different plant distances is particularly interesting and novel. As a key outcome, it would support the application of the polluter-pays-principle. However, the authors do not bring the responsibility question so much into the foreground here, they rather emphasize the opportunity that lies in the development of fair cost sharing mechanisms based on location-specific co-benefits. The authors also investigated the adoption of end-of-pipe technologies to reduce the costs of air pollution caused by coal-fired power plants. They found the so far achieved national emission reduction to be around 70% on average in 2017 and conclude that shifting to non-fossil resources is necessary to exceed this value.

Specific comments and recommendations

As the model behind the analysis has a certain complexity, I recommend to develop a scheme/flow chart to illustrate the different inputs and calculation steps in order to facilitate the reader's comprehension and reproducing by other researchers.

The authors assume that the severity of adverse health effects of coal power plants in a given region, and thus the co-benefits, mainly depends on four factors (110-118): technical aspects of the power plant, population density, baseline mortality rates, and climate conditions. It would be interesting to know how much each factor/sub-factor contributes to the end result, e.g. based on statistical analyses.

The statement that "provinces where the most polluting coal plants are located are not necessarily the provinces that suffered the most" (177-179) is not enough supported by Figure 3. Or in other words, the relationship between polluting and polluted regions as show in the regional comparison in Figure 3 could be made clearer if other options to display the four variables were chosen (e.g. scatter plots, clustering of regions with similar characteristics) and the statistical significance clarified.

Due to the given structure of the article, I recommend to provide an extended and more concrete explanation of important assumptions early (at first occurrence) in the text:

- "we use the capacity-weighted average value to represent the co-benefit of one ton CO₂ emissions reduction in a certain province as well as in the entire country" (156) → compare with "capacity-weighted mean of co-benefit values for all individual plants in that province" (524)
- "The effects of the mega-cities are more prominent when local impacts of air pollution are given

higher weights (Supplementary Fig. 1)" (137) → compare with "in Supplementary Figure 1 we multiply the economic loss within 100km by 3 to take into account the other negative local impacts" (494)

Lines (91-103) in the intro part comprise remarks on methodology and results. I recommend to rephrase in the sense of more describing the research questions/aims of the study here.

I recommend to include meaningful subheadings in the results section to demarcate the individual findings: spatial distribution of health co-benefits, end-of-pipe technologies, and spatial distribution of emissions.

Lines (51-61) I wonder about the full picture here, the other sources for air pollution in China and their significance compared to coal-fired plants? Just as an argument for laying the focus on the energy sector in the paper. I also recommend to add the number of coal plants in China (520) and the number of provinces/grids to clarify the scope of the analysis. In addition, the temporal scope could be more precise in the beginning, I found 2017 (174, 190, 226) and 2015 (459) in different parts of the article.

Lines (59-61) "When resources are constrained, all three strategies need to prioritize certain regions..." The causality of the argument is not clear. What is meant with resources, material or financial resources?

Lines (41-43) "... usually more urgent" evidence/reference missing

Line (98) "some locations" = more distant locations

Lines (360, 365) Zhou and colleagues = Zhou et al.

Suppl. Table 4: title missing

Reviewer #3 (Remarks to the Author):

(see next page)

Report on NCOMMS-20-36232-T. "Location-specific co-benefits of carbon emissions reduction from coal-fired power plants in China"

Summary

This paper estimates location specific benefits of reducing emissions from coal power plants in China. The aim is to allow pollution control and CO₂ policies to be targeted by location since the benefits may vary by 5 times over different locations. Instead of trying to apply atmospheric models over each source, the method involves intake fractions and using only data of population and precipitation around each source to estimate exposure.

Comments

The paper makes a very important point that a ton of SO₂ or NO_x emitted from different locations generates very different damages depending on the surrounding population, winds and rain. The argument that policy makers should recognize these differences in designing control policies is certainly correct. I also commend the authors for noting that the province is the locus of policy making and thus the discussion should be centered at the provincial level. And to note that provinces have different cost pathways to reduce pollution (e.g. different qualities of coal).

I have, however, some difficulties with the overall framework and specific methods used. The first point I like to make is that this is not novel as claimed by the authors. Muller and Mendelsohn (2009) make such calculations and arguments for the U.S. I would also refer the authors to Fowlie and Muller (2019) for a more complex discussion of such location specific policies.

My main puzzle is the choice of discussing this topic in terms of benefits per ton of CO₂ emitted. This may seem to make sense in that reducing coal use will reduce the PM, SO₂ and NO_x emissions at the same time. However, the discussion of policies (page 12) focus on (1) FDG, (2) SCR, (3) bag filters, (4) low-sulfur coal. I am very puzzled how Figure 4 arrives at the benefits of these 4 policies in terms of "per ton CO₂ reduced". These policies reduce PM, SO₂ and NO_x, not CO₂! It seems to be more relevant to focus directly on the benefit of reducing 1 ton of SO₂ or 1 ton of NO_x; to calculate these by location. Then we compare the cost of reducing SO₂ with the benefits of reducing SO₂, etc.

The benefit of reducing 1 ton of CO₂ needs to be compared to the cost of replacing a coal kWh with a non-fossil kWh. This is an important consideration but is not discussed in the policy section of this paper.

The second major point I like to note is the political economy of such a location specific policy in a system of provincial governance. The paper properly notes that damages of the

pollutant emissions occur outside the province in many cases. A national calculation of costs and benefits would differ from the provincial calculation, and thus the central government must design a system of compensation and taxes to deal with this externality. The authors should discuss the complications of such a system.

The valuation of damages is based on incomes in each area. That is, a life saved in Guangdong is considered to be worth more than a life in Guangxi. Such approaches have been very controversial. As an example, there are proposals for US policy-making to use age specific VSL's; these have generated enormous opposition. I think a calculation based on a national valuation should be the main focus; the current location-based valuation could be given as an alternative possibility to show how the maps change using different assumptions.

These calculations of location specific damages are based on intake fractions (IF) estimated from a simple reduced-form regression. If a government were to use this to calculate an externality tax this would likely face opposition from specific plants. Plants would ask for a custom modelling of the dispersion of pollutants from their location and stack height if they expect the estimated damages would be lower than the IF derived one. The implementation of such a policy, I believe, requires a great deal more thought.

I have some questions about the methods. Firstly, intake fractions are not widely known and the first mention on page 5 should have some additional explanation. If I understood the description on page 20, the authors rely on Zhou et al (2006) estimates in Sup Table 2. This study uses only a sample of 29 plants from the early 2000s. It would be good to discuss if this is too small a sample, and that changes in population patterns since then may require a revision. That study only includes precipitation, finer analysis in terms of seasons and winds could be important.

The valuation of damages is based on the US estimate of US\$7mil./person and scaled using an income elasticity of 0.5. This means that a population with $\frac{1}{4}$ of the US income would have a VSL of $\frac{7}{2}$ mil. And a person with $\frac{1}{9}$ of the US income have a VSL of $\frac{7}{3}$ million. The paper does not cite China-based VSL studies, I would say that none have such high valuations. Or more generally, studies of values of total China damages do not use such high numbers.

The paper commented that Beijing and surrounding areas do not have the highest estimated benefits of coal-use reductions but has the strictest control policies and suggest that it is sub-optimal. I think one should note that a prime motivator for the strict control in the 2000's was the run-up to the Beijing Olympics, and many would argue that it was indeed a proper national priority.

The paper suggests (page 18) that it is feasible to integrate carbon pricing and pollution taxes into one unified pricing scheme. I do not think it is so easy, not least since the authors do not cite anyone else with an explicit proposal. As I note above, the carbon price should be compared with the cost of replacing a coal kWh with a non-fossil kWh. This is an additional

complex calculation. Given that FGD and SCR technologies are effective we should probably focus on keeping explicit SO₂ and NO_x prices. The benefit of reducing a ton CO₂ should indeed include the co-benefit of reducing pollution.

Minor points.

1. The reference item 4 is listed as “China Electricity Association ...” Is this correct? Do you mean the China Electricity Council? If it is in Chinese it should be so stated, with the title in pinyin given too.
2. P6. “...from US\$45-236 ...” should have a “per ton CO₂” to make that clear.
3. P8. The provincial averages are computed using capacity weights. Since capacity utilization varies a bit, it would be more accurate to use output (MWh) as weights. Better still would be using fuel consumption as weights.
4. P19. In citing the Global Energy Monitor and the Ctr for Global Dev. as the sources of data, the full references of these sources should be given (book, website, etc). The source of meteorological data should be given.
5. P22. “... Table 2, we calculated ΔC_{ijt} caused ...” I think you mean IFijr

References

Fowlie, Meredith and N. Muller (2019) Market-Based Emissions Regulation When Damages Vary across Sources: What Are the Gains from Differentiation? *Journal of the Association of Environmental and Resource Economists*.

Muller, Nicholas and R. Mendelsohn (2009) Efficient Pollution Regulation: Getting the Prices Right, *American Economic Review*.

Response to reviewers' comments

Location-specific co-benefits of carbon emissions reduction from coal-fired power plants in China

Pu Wang, Cheng-Kuan Lin, Yi Wang, Dachuan Liu, Dunjiang Song, Tong Wu

We have benefitted significantly from the inputs from reviewers, and we sincerely appreciate their suggestions as to how to strengthen the paper. We have made substantial changes to the manuscript according to reviewers' comments. Our response is organized as follows: Each comment is repeated in blue colour, italic texts, followed by a description of the response and the corresponding changes made in the manuscript (in quotes). We also provide a revised manuscript with all changes tracked to facilitate the editors and reviewers' evaluation of our revision. The page and line numbers referred below are the numbers after all the tracked changes are hidden or accepted in the manuscript file.

Reviewer #1 (Remarks to the Author):

This is a good paper analyzing Location-specific co-benefits of carbon emissions reduction from coal-fired power plants in China and therein some economic and non-economic concerns. However, I recommend following changes and/or explanations.

Response: Thank you for your overall comments and the specific suggestions. We find the suggestions very helpful in improving our paper, and we incorporated them into our revision. Though some of your suggestions regarding the format style of the paper are different from the format requirements of the journal, and in these cases, we follow the journal's format guidelines in principle. Please see our detailed response below.

1. The first section heading of Introduction needed to be placed below Abstract.

Response: Thank you for your suggestion. We added a section heading "Introduction" in the manuscript.

2. In the first section provide worth citing references at the end of the sentence with the given addition of fossil fuels share in energy mix: While the effects of carbon emissions

are independent of source locations, the effects of air pollutant emissions are highly dependent on source locations, population distribution, fossil fuels shares in energy mix, climate, and other factors (Shakeel and Zeshan, 2020; Shakeel, 2020).

Response: Thank you for your suggestion. We find the two recommended papers valuable in understanding the interconnection between energy mix change and socio-ecological dynamics. In fact, we think the two papers illustrated the co-benefits of energy structural change in environmental and social domains, particularly in water resources. So we cited the two references at the end of the sentence on Page 4, Line 73:

“Other researchers combined the above methodologies with climate mitigation scenarios, and then assessed the co-benefits of climate policies in air pollution reduction and other environmental issues ¹⁶⁻²¹”.

The two references are #18 and #21 in the revised manuscript, respectively.

3. Though you provided the review in your introduction yet there is need to explain your analytical framework/empirical models with the help of equations and explanation. If you find that you have provided enough information in the methodology then it may be skipped.

Response: Yes, in the Methods section, we explained our models, assumptions, and equations step-by-step, and in addition, we added a flow chart in Supplementary Information (see Supplementary Fig. 1) to illustrate the inputs and calculation steps of the methods that are used in this study.

4. The section of data and variable explanation will also be needed as a separate or sub section of the empirical models/methodology. Please note that in a research article, there are at least five sections namely; Introduction, Review of Literature, Empirical/Theoretical Model and Data description, Results and conclusion. Do provide these in your paper.

Response: Thank you for your suggestion. We understand that in many research papers, it is typical to include the five sections you listed. But the journal *Nature Communications* has very specific format requirements. It requires sections in the order of: Introduction, Results, Discussion, and Methods, et al., and section headings that are not included in its format guidelines are not permitted. So we made relevant changes to the manuscript to comply with the format requirements of the journal. The data description part is included in the Methods section.

5. You provided methodology after the results while it should before or after the theoretical model and data description section as elaborated before.

Response: Please see our response to point #4. We made relevant changes to comply with the format requirements of the journal.

6. Do provide the conclusion of the study as indicated in point no. 4 above.

Response: We made relevant changes to comply with the format requirements of the journal. We also added additional discussion and conclusions in the revised manuscript, presented below:

On Page 22, starting from Line 415, we wrote:

“A series of challenges need to be addressed in order to better integrate our analysis into actual policies. First, the current models still have large uncertainties in assessing exposure and health losses, which can undermine the benefits of location-specific policies ¹. Even state-of-the-art models such as GEOS-Chem also show significant discrepancies between predicted and observed data. Thus, model robustness needs to be further improved to maximize the benefits of precise policy making. Second, we recognize that this study, as well as previous studies on co-benefits, only take into account mortality caused by increase in PM_{2.5} concentration, and does not consider other types of damages, such as higher morbidity and lower work productivity. This problem can be addressed by adding more comprehensive valuation analyses in future models. Third, a series of practical questions need to be addressed during policy implementation, including how often the co-benefits valuation should be updated based on climate, demographic, and epidemiological changes, how to treat plants close to provincial boundaries that are charged differently, and how to prevent rent-seeking in the process of assigning different prices to different plants. This type of issue is not unique for the policies proposed in this study, and can be addressed by administrative discretion and referring to precedents. And fourth, policy makers need to make various political and ethical decisions about valuation parameter choices and fairness. Choosing different benchmark values of statistical life (VSLs) will significantly change the values of co-benefits, thus affecting the stringency of coal restriction policies (Supplementary Figure 4); Adjusting VSLs based on income levels in different regions can generate more precise locational estimations, but faces significant ethical challenges (Supplementary Figures 2&3); Socio-political considerations can outweigh climate and environmental concerns; for instance, as the national capital city, Beijing has the

strictest coal control policies, even though it does not have the highest estimated co-benefits. Such value-based decisions are beyond the scope of our analysis.

Overall, the methods developed in this study represent a scientifically sound approach to connecting facility-level information to macro-level policy-making by means of data-intensive analysis. Going forward, it is possible to modify these methods and apply them to similar issues, such as assessing the locational impacts of power systems on water resources and land use, thereby capacitating more cost-effective environmental policies with spatial nuances in mind.”

7. Similarly check for spelling/grammar mistakes and also explain why CO2 equivalence were not used because there are also other gases like chloropholoro carbons, methane gas etcetera and thus CO2e would be a better estimate of pollution instead of CO2? You can use footnote for this explanation.

Response: Thank you for your suggestion. We asked a native English speaker who is also a researcher in our field to closely check and correct the spelling and grammar issues throughout the manuscript. As to the issue of CO₂ v.s. CO₂ equivalent, because our research is focused on coal-fired power plants, and CO₂ is the only significant greenhouse gas emitted during the process of electricity generation. The emissions of other greenhouse gases, such as methane, are negligible. That’s why we chose CO₂, rather than CO₂ equivalent, in our calculations. Similar studies on coal-fired power plants, such as Li et al. 2018 ² and Cai et al. 2018 ³, also used CO₂, rather than CO₂ equivalent. Had we analysed natural gas-fired power generation, methane leakage would become a major concern, and the use of CO₂ equivalent would be more appropriate.

References

Zeshan, M. Shakeel, M. (2020) Adaptations and Mitigation Policies to Climate Change: A Dynamic CGE-WE Model. Singapore Economic Review.

DOI: <https://doi.org/10.1142/S0217590820500654>

Shakeel, M. (2020) Economic output, export, fossil fuels, non-fossil fuels and energy conservation: evidence from structural break models with VECMs in South Asia. Environmental Science and Pollution Research. <https://doi.org/10.1007/s11356-020-10729-9>

Response: We cited these two studies in appropriate places in the manuscript. See our response to Point #2.

Reviewer #2 (Remarks to the Author):

General comments

The article investigates the location-specific health co-benefits of policy measures implemented at the provincial level to reduce carbon emissions from coal-fired power plants in China. Health co-benefits are given in USD per-ton CO₂ emissions reduction and represented in maps and therefore form a sound information basis for political decision makers (compared to otherwise abstract physical metrics).

By means of the applied methodology it is possible to identify areas (and even individual plants, however, with regional-grid-average technology specifications) with a high potential for generating co-benefits resulting from investments in carbon mitigation. A strength of the article is the interdisciplinary approach, the combination of economic, ecological and social indicators to evaluate the issue. The authors integrated state-of-the-art concepts for their holistic investigation. Selected input data are actual and representative for the case study.

Concerning the results, the shown interrelationship between air pollutant-specific costs and different plant distances is particularly interesting and novel. As a key outcome, it would support the application of the polluter-pays-principle. However, the authors do not bring the responsibility question so much into the foreground here, they rather emphasize the opportunity that lies in the development of fair cost sharing mechanisms based on location-specific co-benefits. The authors also investigated the adoption of end-of-pipe technologies to reduce the costs of air pollution caused by coal-fired power plants. They found the so far achieved national emission reduction to be around 70% on average in 2017 and conclude that shifting to non-fossil resources is necessary to exceed this value.

Response: Thank you for the very insightful comments and recommendations. We addressed your concerns point-by-point below, and integrated your recommendations into our revision, which help significantly improve our manuscript. As to the application of the polluter-pays-principle, it was implied in our original manuscript through the proposed pollution tax that let the polluters bear the external costs. In the revised manuscript, we explicitly argue that our outcome would support the application of the polluter-pays-principle through a pollution tax, though compensation and cost-sharing mechanisms are also important in helping the less developed regions reduce emissions. On Page 16, Line 307, we wrote:

“This indicates that in less populated regions (also the less developed regions) such as the northwest, while the costs of pollutant removal are usually borne locally, the adoption of such technologies largely benefit other provinces. Therefore, besides a strict application of the polluter-pays-principle by imposing a pollution tax equal to damage, it is necessary to design compensation and cost-sharing mechanisms to encourage the adoption of pollution reduction technologies in less developed regions.”

In addition, we added more discussion on the complexity of such policies in practice in the Discussion Section.

Specific comments and recommendations

As the model behind the analysis has a certain complexity, I recommend to develop a scheme/flow chart to illustrate the different inputs and calculation steps in order to facilitate the reader ' s comprehension and reproducing by other researchers.

Response: Thank you for the valuable suggestion. We made a flow chart that includes model inputs, calculation steps, and outputs, and included the flow chart in Supplementary Information as Supplementary Fig. 1.

Supplementary Fig. 1 Flow chart of model inputs, calculation steps, and outputs.

The authors assume that the severity of adverse health effects of coal power plants in a given region, and thus the co-benefits, mainly depends on four factors (110-118): technical aspects of the power plant, population density, baseline mortality rates, and climate conditions. It would be interesting to know how much each factor/sub-factor contributes to the end result, e.g. based on statistical analyses.

Response: For each specific plant, the relative importance of the four factors can vary significantly. To understand the relative importance of the four factors on average, we conducted multiple simulation analyses to assess the average effects of a 10% increase in each of the four factors on the co-benefit of per ton CO₂ emissions reduction. The results of the analyses, as well as the interpretation of the results, are added to the following places in the main text and supplementary information.

In main text Page 6, lines 129, we added:

“For each specific plant, the relative importance of the four factors can vary significantly. We conduct multiple simulations to assess the relative importance of each factor on average (supplementary note 1). Overall, population density has the dominant impact on co-benefit values, and precipitation and technology specifications (represented by emission factors) have modest impacts, while baseline mortality rate change has a relatively small impact.”

In Supplementary Information Page 2, we added:

“The four major factors that influence the value of locational health co-benefits are: 1) the technological specifications of a power plant; 2) population within different ranges from a plant; 3) baseline mortality rates of diseases related to air pollution, including ischemic heart disease (IHD), chronic obstructive pulmonary disease (COPD), and lung cancer; and 4) local climatic conditions, most importantly precipitation levels.

For each specific plant, the relative importance of the four factors can vary significantly. To understand the relative importance of the four factors on average, we ran multiple simulations to evaluate the national average effects of a 10% increase in each factor. (1) A 10% increase in population density within 500km from a power plant will increase the co-benefit value by 4.17% on average; since population size within 500km of a power plant ranges from 1.4 million to 443 million, population density change is a dominant factor that affects co-benefit value. (2) A 10% increase in annual precipitation will decrease the co-benefit value by roughly 3%, while annual precipitation in China ranges from 30mm to 2770mm. (3) A 10% increase in the emission factors of SO₂, NO_x, and PM_{2.5} simultaneously will increase the co-benefit value by 10%, while a 10% increase in CO₂ emission factors will decrease the co-benefit value by 9.09%; changes in emission factors range between 100%-344%. (4) A 10% increase in baseline mortality rates of the three relevant diseases in the province where a power plant is located will increase the co-benefit value by 0.51%; changes in baseline mortality rates range between 100% to 232%. In summary, population density has the dominant impact on co-benefit values, and precipitation and technology specifications (represented by emission factors) have modest impacts on co-benefit values, while baseline mortality rate change has relatively small impact.”

The statement that “provinces where the most polluting coal plants are located are not necessarily the provinces that suffered the most ” (177-179) is not enough supported by Figure 3. Or in other words, the relationship between polluting and polluted regions as show in the regional comparison in Figure 3 could be made clearer if other options to display the four variables were chosen (e.g. scatter plots, clustering of regions with similar characteristics) and the statistical significance clarified.

Response: Thank you for the suggestion. We changed the way of presentation in Figure 3 to more clearly display the differences between polluting and polluted regions. In the revised Figure 3, Figure 3a and 3c show the damage of per ton CO₂ emission and the damage of annual total emissions, respectively, from coal-fired power plants located in each province; these two calculations are based on the damage caused by polluting sources. Figure 3b and 3d present the annual per capita loss and the total loss suffered by each province due to coal-fired power generation nationwide. As Figure 3 shows, the rankings of damage are very different from the rankings of losses. The provinces connected with blue lines have power plants that are relatively more damaging, but these provinces suffer relatively less. In contrast, provinces connected by red lines have relatively less damaging power plants, but these provinces suffer more from coal-fired power generation. The orders of the total damages and losses in Figures c and d show similar mismatches. Thus, the revised Figure 3 can more clearly illustrate our point that “provinces where the most polluting coal plants are located are not necessarily the provinces that suffered the most”, and highlights the need to target more polluting regions, rather than the regions that suffer more losses.

Figure 3. The difference between health cost valuations based on emission sources vs. the affected regions. (a) Value of health co-benefit from reducing one ton CO₂ emissions from coal-fired power plants located in each province; (b) Per capita health loss suffered by each province due to air pollution caused by all coal plants in China in 2017; (c) Total value of health loss caused by coal plants within each province; (d) Total value of health loss suffered by each province due to air pollution caused by all coal plants in China in 2017. Provinces connected by blue lines have power plants that are relatively more damaging, but these provinces suffer relatively less; provinces connected by red lines have relatively less damaging power plants, but these provinces suffer more from coal-fired power generation.

Relevant changes in Main text Page 10, Line 194:

“While Figures 1 and 2 present the damage caused by power plants located in different regions, Supplementary Figure 5 presents the economic losses suffered at different locations due to air pollution from all power plants. The annual national loss added up to 460.1 billion USD in 2017, or around 3.7% of the country’s total GDP in the same year. Figure 3 illustrates the difference between the rationales behind Figures 1&2 and Supplementary Figure 5. Figures 3a and 3c show the co-benefit of per ton CO₂ emission

and the damage of annual total emissions, respectively, from coal-fired plants located in each province; Figures 3b and 3d present the annual per capita loss and the total loss suffered by each province due to coal-fired power generation nationwide. As Figure 3 shows, the provinces where the most polluting coal plants (inducing the highest losses) are located (Figures 3a and 3c) are not necessarily the provinces that suffered the most (Figures 3b and 3d). For instance, power plants in Hubei and Chongqing rank the second and the fourth in terms of the average damage, but their per capita losses rank 10th and 14th, respectively. In contrast, power plants in Anhui cause relatively small damages (ranks 20th nationally), but per capita loss in Anhui ranks the 6th. The orders of the total damages and losses in Figures 3c and 3d show similar mismatches. The majority of previous studies on health costs of air pollution addressed the latter, namely estimating the losses suffered by affected regions²⁻⁴. However, from the perspective of energy policy targeting, it is more important to address the emission sources that induce the highest costs. Thus, it is arguable that results in Figures 1&2 have more direct policy interpretations in energy planning and siting, as well as in carbon pricing and environmental tax policies. ”

Due to the given structure of the article, I recommend to provide an extended and more concrete explanation of important assumptions early (at first occurrence) in the text:

- *“we use the capacity-weighted average value to represent the co-benefit of one ton CO2 emissions reduction in a certain province as well as in the entire country ” (156) → compare with “capacity-weighted mean of co-benefit values for all individual plants in that province ” (524)*
- *“The effects of the mega-cities are more prominent when local impacts of air pollution are given higher weights (Supplementary Fig. 1) ” (137) → compare with “in Supplementary Figure 1 we multiply the economic loss within 100km by 3 to take into account the other negative local impacts ” (494)*

Response: Thank you for the suggestion. We have made relevant changes in the following places to ensure that our key assumptions are explained before presenting the corresponding results:

On Page 9, Line 177:

“Thus, we use capacity-weighted mean of co-benefit values for all individual plants in a province to represent the co-benefit value in that province (Fig. 2), and use the same method to calculate the national co-benefit.”

On Page 8, Line 158:

“We analyse how the pattern in Figure 1 changes under different assumptions in Supplementary notes 2&3. First, we re-estimate co-benefits using VSLs based on GDP per capita at each city, in contrast to a universal national VSL used in Figure 1. Second, a power plant can have many detrimental effects on adjacent areas other than PM_{2.5} (e.g., coal ash and heavy metal deposition). We multiply the economic loss within 100km by three to take into account the other negative local impacts (see methods). Third, we re-estimate co-benefits using different national VSLs based on literature review. The corresponding results are presented in Supplementary Figures 2,3, and 4.”

Lines (91-103) in the intro part comprise remarks on methodology and results. I recommend to rephrase in the sense of more describing the research questions/aims of the study here.

Response: Thank you for the suggestion. We made some changes the paragraph and moved some sentences to the Results section. Though the journal’s format instructions require that we discuss the results and significance of the study in the last paragraph of Introduction. So we keep some contents in this paragraph to comply with the requirement. The revised paragraph is shown on Page 5, Line 92:

“To address this gap, we establish an interdisciplinary approach to assess facility level co-benefits at specific locations. Our model takes advantage of a complete coal plants dataset, the latest fine-scale demographic and health data, and air pollution exposure estimations calibrated specifically for coal plants in China. Our analyses capture technological features, fuel quality, and the specific location of each individual power plant, as well as the relevant demographic, economic, weather, and epidemiological information of affected regions. We find that co-benefits in “hotspot” locations can be up to five times higher than those in more distant locations, and that provinces should use different techno-economic strategies to reduce pollutant emissions. These high-resolution results have straight-forward policy interpretations, and can significantly improve the design of China’s emissions trading system, environmental pollution taxes, and other policies aimed at changing the country’s fossil-fuel-dominant energy structure.”

I recommend to include meaningful subheadings in the results section to demarcate the individual findings: spatial distribution of health co-benefits, end-of-pipe technologies, and spatial distribution of emissions.

Response: Thank you for the suggestion. We added five subheadings in the Results section, namely “Spatial distribution of co-benefits”, “Co-benefits at provincial level”, “Impacts of end-of-pipe technologies on co-benefits”, “Analysis of pollutant-specific co-benefits”, and “Comparison of results based on different models”.

Lines (51-61) I wonder about the full picture here, the other sources for air pollution in China and their significance compared to coal-fired plants? Just as an argument for laying the focus on the energy sector in the paper. I also recommend to add the number of coal plants in China (520) and the number of provinces/grids to clarify the scope of the analysis. In addition, the temporal scope could be more precise in the beginning, I found 2017 (174, 190, 226) and 2015 (459) in different parts of the article.

Response: In the revised manuscript, we added the specific percentages of national SO₂, NO_x, and primary PM_{2.5} emissions attributable to coal-fired power generation to illustrate the importance of power sector in air pollution control. We also added the approximate number of coal-fired power plants (over 2,000). But we do not have the exact number, due to incomplete inventory data and construction and retirement of plants overtime. Our dataset included 2475 power plants (for some plants each generation unit is a single entry), which represented 93% of China’s coal power capacity in 2017. Concerning the word limit for the Introduction section according to the journal’s format instructions, we did not include the description of air pollution sources in other sectors; instead, we provided the reference to Zheng et al. 2018, which has detailed information on China’s sectoral air pollutants emissions. We also mentioned the number of provinces/grids in the Introduction and Results sections.

Also, in the revised manuscript, we replaced the 2015 GDP data with 2017 GDP data, and now all the key analyses are based on 2017 data. The relevant changes in the manuscript are presented below:

On Page 3, Line 51:

“The electric power sector is the largest source of carbon emissions in China, accounting for roughly half of the country’s total carbon emissions^{5,6}. There were over 2,000 coal-fired power plants in Mainland China’s 31 provinces in 2017, with generation capacity of 980 GW and electricity generation of 4.1 trillion kWh, accounting for 55% of total capacity and 65% of total generation, respectively⁷. Meanwhile, coal-fired plants have been a major cause of the country’s serious air pollution in recent years^{8,9}, contributing to 17%, 19%, and 8% of China’s total SO₂,

NO_x, and primary PM_{2.5} emissions respectively in 2017 ¹⁰.”

On Page 24, Line 464:

“Population distribution data are based on the 1x1 km global population grid data for 2017 developed by Oak Ridge National Laboratory’s LandScan project, which is the finest-grain population distribution data to date. China’s national and prefecture-level GDP data were extracted from Chinese statistical yearbooks for the year 2017.”

On Page 29, Line 575:

“We used the VSL suggested by the US EPA ¹¹ as the baseline (VSL_{base}), which is \$7.4 million per person in 2006 value, and adjusted it to 2017 value.”

Lines (59-61) “When resources are constrained, all three strategies need to prioritize certain regions…” The causality of the argument is not clear. What is meant with resources, material or financial resources?

Response: We clarified in the text that we refer to the financial and administrative resources, both in public and private sectors, that are needed for mitigating the effects of coal-fired power plants. Financially, the government needs to provide compensations, tax breaks, or other incentives to encourage power plants to retrofit, relocate, or shutdown; the power plants owners also face potential economic losses in retrofitting or early closure of their plants. The administrative resources that can be allocated to power plants retrofitting, relocation, closure, and financial compensation are also limited. So it is important to prioritize regions that need these resources the most. We made relevant change in the text as below:

On Page 3, Line 58:

“In response, China has adopted three major strategies to reduce the environmental impacts of power plants: eliminating plants with outdated technologies, ultralow emissions retrofitting, and optimizing the siting of plants, and substantial financial and administrative resources have been invested by plant owners and the government ¹²⁻¹⁴. When these resources are constrained, all three strategies need to prioritize certain regions, and this prioritization requires location-specific social and economic justifications, which is the goal of this study.”

Lines (41-43) “... usually more urgent ” evidence/reference missing

Response: It’s true that there is urgent need to address both global climate change and local air pollution, as the time window to avoid catastrophic climate effects is closing. But relative to developed nations that have already solved air pollution problems, developing countries, such as China and India, still face severe air pollution challenges. While the effects of climate change mostly affect future generations (though the short term effects are emerging in recent years), air pollution is causing millions of premature deaths each year in China and India right now. Our intention is by no means downplaying the urgency of climate change mitigation. Instead, we argue that integrating air pollution control targets with climate change mitigation will more effectively motivate developing countries to take actions on climate change now. Due to the word limit, we cannot elaborate on this point in the Introduction. So we provided two references on Page 2, Line 41 to support this point. One is an analysis about annual premature deaths caused by air pollution, particularly in China and India (Reference #2), the other is the time scale of the effects of climate change (Reference #1).

Line (98) “some locations ” = more distant locations

Response: We changed the sentence on Page 5, Line 98 to:

“We find that co-benefits in “hotspot” locations can be up to five times higher than those in more distant locations, and that provinces should use different techno-economic strategies to reduce pollutant emissions.”

Lines (360, 365) Zhou and colleagues = Zhou et al.

Response: We made the changes in the text on Page 25, Line 478 and Line 483.

Suppl. Table 4: title missing

Response: We added a title for Supplementary Table 4: “Changes in national emission factors from 2010 to 2017”.

Reviewer #3 (Remarks to the Author):

Report on NCOMMS-20-36232-T. "Location-specific co-benefits of carbon emissions reduction from coal-fired power plants in China"

Summary

This paper estimate location specific benefits of reducing emissions from coal power plants in China. The aim is to allow pollution control and CO2 policies to be targeted by location since the benefits may vary by 5 times over different locations. Instead of trying to apply atmospheric models over each source, the method involves intake fractions and using only data of population and precipitation around each source to estimate exposure.

Comments

The paper makes a very important point that a ton of SO2 or NOx emitted from different locations generates very different damages depending on the surrounding population, winds and rain. The argument that policy makers should recognize these differences in designing control policies is certainly correct. I also commend the authors for noting that the province is the locus of policy making and thus the discussion should be centered at the provincial level. And to note that provinces have different cost pathways to reduce pollution (e.g. different qualities of coal).

I have, however, some difficulties with the overall framework and specific methods used. The first point I like to make is that this is not novel as claimed by the authors. Muller and Mendelsohn (2009) make such calculations and arguments for the U.S. I would also refer the authors to Fowlie and Muller (2019) for a more complex discussion of such location specific policies.

Response: Thank you for the very insightful comments. The purpose of our paper is indeed to help improve China's climate and energy policy making in terms of better geographic targeting and prioritization. From this perspective, the Muller and Mendelsohn (2009) paper is trailblazing and has provided important theoretical and methodological inspirations for our study. Also, the Fowlie and Muller (2019) paper discusses the cost-benefit uncertainties in policy differentiation, and helps us make more realistic discussions in our policy proposals. We have included these two key references in the manuscript, and have added additional discussion in relevant sections, presented in our responses below.

We also would like to point out that the rationale of our study is not exactly the same

as that of the Muller and Mendelsohn (2009) and Fowlie and Muller (2019) papers. The latter two papers focus directly on the damages of SO₂ and NO_x emissions. In contrast, our paper highlights the health co-benefits of carbon emission mitigation policies, which is a key concept in recent literature (e.g. Shindell 2012¹⁵, West 2013⁴, Lelieveld 2015¹⁶, Li 2018²) to provide justification and motivation (in addition to climate itself) for developing countries to accelerate climate mitigation. Our study develops scientific methods to more accurately estimate location-specific co-benefits, and explores means to make implementable policies based on location-specific co-benefits. From this perspective, our study builds upon Muller and Mendelsohn (2009) and other studies, and makes new contributions to this field. We removed the word “novel” on Page 5 Line 92 to show respect to the earlier work that our study builds upon, and to comply with the journal’s editorial policy.

My main puzzle is the choice of discussing this topic in terms of benefits per ton of CO₂ emitted. This may seem to make sense in that reducing coal use will reduce the PM, SO₂ and NO_x emissions at the same time. However, the discussion of policies (page 12) focus on (1) FDG, (2) SCR, (3) bag filters, (4) low-sulfur coal. I am very puzzled how Figure 4 arrives at the benefits of these 4 policies in terms of “per ton CO₂ reduced”. These policies reduce PM, SO₂ and NO_x, not CO₂! It seems to be me to be more relevant to focus directly on the benefit of reducing 1 ton of SO₂ or 1 ton of NO_x; to calculate these by location. Then we compare the cost of reducing SO₂ with the benefits of reducing SO₂, etc.

The benefit of reducing 1 ton of CO₂ needs to be compared to the cost of replacing a coal kWh with a non-fossil kWh. This is an important consideration but is not discussed in the policy section of this paper.

Response: Thank you for the comment. As we explained in our response to the previous point, our study is focused on the co-benefits of carbon emissions reduction, rather than air pollution reduction per se. When calculating the benefits of carbon reduction, we should not only estimate the benefits of climate mitigation, which is a global public good, but should also take into account the “by-product” benefits of local air pollutants reduction. This is particularly important to motivate countries that suffer from severe air pollution problems, such as China and India, to reduce the consumption of fossil fuels. We revised the first paragraph in Introduction to more clearly highlight this point.

The value of co-benefit per ton CO₂ reduction is determined by the ratio of CO₂ and air pollutants emitted from per unit electricity generation. Applying end-of-pipe removal

technologies, such as FDG, SCR, and bag filters, will significantly change the carbon-air pollutants ratio, thus will affect the co-benefit value. After discussing the geographic distribution of co-benefits based on grid-average technological deployment rates (Figures 1 and 2), we need to discuss how different penetration rates of FDG, SCR, and, bag filter technologies will change the co-benefit value (Figure 4).

From a policy making perspective, the results in Figures 1 and 2 are not fixed and should be updated regularly (i.g. annual update) based on changes in technological specifications in provinces. A province can deploy more end-of-pipe control technologies to change the carbon-air pollutants ratio, thus can change its relative priority in national coal plants phasing out. It can also receive a more favourable trading ratio in a permit trading system, or a more lenient tax rate in a tax system. Our calculations in Figure 4 can provide basis for such updates.

We agree with you that we need to better explain our rationale of the focus on co-benefits, and that in Figure 4 it is more straight forward for readers to first obtain the benefits of reducing one unit SO₂, NO_x or primary PM_{2.5} at different locations. We revised Figure 4 to first present the benefits of SO₂, NO_x, and PM_{2.5} reduction, and then convert the above values to co-benefits of per ton CO₂ reduction based on the carbon-air pollutants emission ratios, and we explain why we made such conversions.

We also added discussion on the comparison of the benefit of reducing CO₂ with the cost of replacing coal power with renewable energy. We cited two studies, one on the costs of renewable energy, conducted by the International Energy Agency, the other on the cost of coal power in China, to show that the levelized cost of electricity (LCOE) of coal power is still lower than that of wind and solar power in China. Thus, adding a carbon price and air pollution costs to coal power price is important to help enhance the relative market competitiveness of wind and solar power, and to reduce the total coal consumption in power sector. Though we did not have extensive discussion on the comparison between coal power and renewable energy, as it is not the primary focus of our study.

The corresponding changes in the manuscript are listed below:

Page 2, Line 39:

“Developing countries, such as China and India, face the dual challenge of climate change and air pollution, with the latter usually more urgent and directly associated with domestic welfare^{16,17}. Therefore, integrating air pollution control into climate

governance can be a highly effective means to motivate developing countries to accelerate climate change mitigation¹⁸. Reducing CO₂ emissions from electric power generation will simultaneously reduce the emissions of sulphur dioxide (SO₂), nitrogen oxides (NO_x), and primary PM_{2.5}. While the effects of carbon emissions are independent of source locations, the effects of air pollutant emissions are highly dependent on source locations, population distribution, climate, and other factors^{1,19}. Thus, spatially nuanced policies are key to maximizing the achievement of co-benefits.”

Page 13, Line 231:

“A power plant can adopt different methods to reduce the emissions of CO₂, SO₂, NO_x, and particulate matters, and the health co-benefits calculation in Figure 1 can be adjusted based on the plant’s technological choices. Specifically, the value of co-benefit is determined by the ratio of carbon and air pollutants emitted from per unit electricity generation. Applying end-of-pipe removal technologies that reduce emissions of SO₂, NO_x, and PM_{2.5} will significantly change the carbon-air pollutants ratio, thus will affect the co-benefits value. Therefore, the results in Figures 1 and 2 are not fixed and should be updated regularly (e.g. annual update) based on changes in technological specifications in provinces. Calculations in Figure 4 provide basis for such updates.

We analyse the potential benefits of four representative technologies in reducing air pollutants emissions in each province, first in terms of the benefits per kWh electricity production (Figure 4a), and then we convert the values into health co-benefits per ton CO₂ emissions reduction based on carbon-air pollutants ratios (Figure 4b).”

Page 14, Line 268:

“...we find the national emissions reduction due to end-of-pipe control technologies was around 70% on average in 2017... Thus, to achieve deeper reductions in CO₂ and air pollutants emissions, it is necessary to reduce the total amount of coal use in power generation. Since the levelized cost of electricity (LCOE) of coal power is still lower than that of wind and solar power in China^{20,21}, adding a carbon price and air pollution costs to coal power price is important to help enhance the relative market competitiveness of wind and solar power, and to reduce the total coal consumption in power sector.”

Revised Figure 4:

Figure 4. Potential benefits of emission reduction technologies and the impacts on co-benefit values. a. Potential benefits of four emission reduction methods in reducing health costs per kWh electricity generation. b. Impacts of four emission reduction methods on co-benefit values per ton CO₂ emission. The four representative methods to reduce emissions are (1) flue gas desulfurization (FGD); (2) selective catalytic reduction (SCR); (3) switching from electrostatic precipitators to fabric filters (FAB); and (4) switching from high-sulfur coal to coal with the national average sulfur content. “Residue” represents the value under the ideal situation when all methods are perfectly adopted. The black and yellow diamond-shaped dots represent the actual provincial values in 2017.

The second major point I like to note is the political economy of such a location specific policy in a system of provincial governance. The paper properly notes that damages of the pollutant emissions occur outside the province in many cases. A national calculation of costs and benefits would differ from the provincial calculation, and thus the central government must design a system of compensation and taxes to deal with this externality. The authors should discuss the complications of such a system.

Response: Thank you for the suggestion. We think there are two types of complications in terms of the location-specific policy. The first type involves the two points you raised below: the choice of valuation approaches for provinces and the robustness of the intake fraction method, which are about the inter-provincial fairness and scientific validity of assumptions and models used for the calculation. We address these two issues in response to your next two comment points correspondingly. The second type involves

how to treat the real-world challenges during policy implementation as well as relevant fairness and ethical issues in the design of national policies. In the Discussion section, we added a paragraph to discuss four types of such challenges. The relevant changes in the manuscript are on Page 22, starting from Line 415:

“A series of challenges need to be addressed in order to better integrate our analysis into actual policies. First, the current models still have large uncertainties in assessing exposure and health losses, which can undermine the benefits of location-specific policies ¹. Even state-of-the-art models such as GEOS-Chem also show significant discrepancies between predicted and observed data. Thus, model robustness needs to be further improved to maximize the benefits of precise policy making. Second, we recognize that this study, as well as previous studies on co-benefits, only take into account mortality caused by increase in PM_{2.5} concentration, and does not consider other types of damages, such as higher morbidity and lower work productivity. This problem can be addressed by adding more comprehensive valuation analyses in future models. Third, a series of practical questions need to be addressed during policy implementation, including how often the co-benefits valuation should be updated based on climate, demographic, and epidemiological changes, how to treat plants close to provincial boundaries that are charged differently, and how to prevent rent-seeking in the process of assigning different prices to different plants. This type of issue is not unique for the policies proposed in this study, and can be addressed by administrative discretion and referring to precedents. And fourth, policy makers need to make various political and ethical decisions about valuation parameter choices and fairness concerns. Choosing different benchmark values of statistical life (VSLs) will significantly change the values of co-benefits, thus affecting the stringency of coal restriction policies (Supplementary Figure 4); Adjusting VSLs based on income levels in different regions can generate more precise locational estimations, but faces significant ethical challenges (Supplementary Figures 2&3); Socio-political considerations can outweigh climate and environmental concerns; for instance, as the national capital city, Beijing has the strictest coal control policies, even though it does not have the highest estimated co-benefits. Such social and political decisions are beyond the scope of our analysis, though our calculation based on environmental benefits can serve as a benchmark for actual policy making.”

The valuation of damages is based on incomes in each area. That is, a life saved in Guangdong is considered to be worth more than a life in Guangxi. Such approaches have been very controversial. As an example, there are proposals for US policy-making to use age specific VSL's; these have generated enormous opposition. I think a

calculation based on a national valuation should be the main focus; the current location-based valuation could be given as an alternative possibility to show how the maps change using different assumptions.

Response: Thank you for the suggestion. We accepted your suggestion and used a VSL based on national average per capita GDP to calculate the co-benefit values in the main text, and put the results based on local VSLs in the supplementary information (Supplementary Fig. 2 and Supplementary Fig. 4c) for comparison. The updated Figure 1 and Figure 2 in the main text now reflect the calculation based on national VSL. Correspondingly, we modified the description of calculations in the main text and SI. The relevant changes are shown below:

On Page 8, Line 158:

“We analyse how the pattern in Figure 1 changes under different assumptions in Supplementary notes 2&3. First, we re-estimate co-benefits using VSLs based on GDP per capita at each city, in contrast to a universal national VSL used in Figure 1.”

On Page 3 in Supplementary Information:

“Supplementary note 2:

The results in Figure 1 are based on a universal value of statistical life (VSL) calculated from China’s national per capita GDP in 2017. The locational co-benefit values will change with different assumptions. First, we re-estimate the co-benefits using VSLs based on GDP per capita at each of China’s 337 prefecture-level cities (Supplementary Fig. 2). Compared to results in Figure 1, the co-benefit values in high income regions (such as Beijing, Tianjin, and Shanghai) increase slightly, while the values in low income regions (such as Gansu, Guangxi, and Yunnan) decrease slightly. But overall, the patterns in the two figures are not significantly different from each other.”

Revised Figure 1 based on a universal national VSL:

Figure 1. Health co-benefits of per-ton CO₂ emissions reduction from coal-fired power plants in different locations (Unit is in USD).

Revised Figure 2 based on a universal national VSL:

Figure 2. Capacity-weighted average health co-benefits of per-ton CO₂ emissions reduction in different provinces. (Unit is in USD)

Newly added Supplementary Fig. 2 based on city-specific VSLs:

Supplementary Fig. 2 Health co-benefits of per-ton CO₂ emissions reduction based on GDP per capita at prefecture-level. (Unit is in USD).

Newly added Supplementary Fig. 4 (c) based on city specific VSLs:

Supplementary Fig. 4 Provincial co-benefits based on alternative VSLs and elasticities. (Unit is in USD); c. prefecture-level VSLs based on local income, using the USEPA recommended benchmark VSL and an elasticity of 0.5;

These calculations of location specific damages are based on intake fractions (IF) estimated from a simple reduced-form regression. If a government were to use this to calculate an externality tax this would likely face opposition from specific plants. Plants would ask for a custom modelling of the dispersion of pollutants from their location and stack height if they expect the estimated damages would be lower than the IF derived one. The implementation of such a policy, I believe, requires a great deal more thought.

Response: Thank you for raising this important point. When choosing our methods, we face the trade-off between state-of-the-art air quality models that generate fine-scale results for each power plant but are very time-consuming, such as the GEOS-Chem model, and less data intensive models such as the intake fractions (IF) method that are more time-efficient to run 2000 plus simulations for all the power plants in China and are reliable for large-scale analysis.

Since the goal of this study is to improve China's national and provincial level policies on coal-fired power plants, rather than focusing on the impacts of a single power plant, we think it is more important for us to use a model that is time-efficient to process a large number of simulations. That is why we regard the intake fraction method as a good match to our need.

We also agree it is very important to ensure that the model results are reliable at large geographic scale. So in the revised manuscript, we added a section to use GEOS-Chem model to statistically test the accuracy of the IF method. We find that the results of IF model are tightly correlated with results from GEOS-Chem model (Figures 6&7 and Supplementary note 4). More detailed descriptions of model comparison are provided in the response to the next comment point.

As you pointed out, if the government wants to calculate an externality tax for an individual plant, then the IF model is not customized enough, and a much sophisticated model, such as GEOS-Chem, should be used. We made relevant changes to the manuscript to discuss this issue, on Page 18, Line 347:

“Thus, overall, we think that the IF model has the advantage of time efficiency in processing a large number of simulations and its results are reliable for policy analysis at large geographic scales. If the government wants to calculate an externality tax for an individual plant based on more comprehensive variables, then the IF model is not sufficiently customized, and a more sophisticated model, such as GEOS-Chem, should be used.”

I have some questions about the methods. Firstly, intake fractions are not widely known and the first mention on page 5 should have some additional explanation. If I understood the description on page 20, the authors rely on Zhou et al (2006) estimates in Sup Table 2. This study uses only a sample of 29 plants from the early 2000s. It would be good to discuss if this is too small a sample, and that changes in population patterns since then may require a revision. That study only includes precipitation, finer analysis in terms of seasons and winds could be important.

Response: Thank you for the suggestion. We added the explanation of the intake fractions (IF) method at its first occurrence on Page 5, Line 108:

“We obtained the locational health co-benefits of per ton carbon emissions reduction in four major steps (see details in Methods and Supplementary Figure 1): 1) using an intake fractions model to estimate population exposure to air pollutants from power plants; an intake fraction is defined as the fraction of material or its precursor released from a source that is eventually inhaled by a population;”

We then address your concerns about the accuracy of the IF method from two perspectives. First, we added more clarifications to explain that the Zhou et al. 2006 study used data that are actual and representative, in Supplementary Note 4:

“Zhou et al. 2006²² study used data that are actual and representative, and their model coefficients were calibrated specifically for China’s coal-fired power plants nationwide. They randomly selected one power plant located in each of 29 provincial administrative units in Mainland China. Thus these plants were distributed roughly evenly across the country and covered all the six regional grids. The model they used was CALPUFF, which is a multi-layer, multi-species non-steady-state air quality modelling system that simulates the effects of meteorological conditions on pollutant transport, transformation and removal (<http://www.src.com/>).

While the CALPUFF model takes into account seasonal changes of precipitation and winds, Zhou et al decided to choose population density and precipitation as independent variables to calculate IF coefficients, because they found that population density and precipitation have the strongest predictive power and can explain most of variations in exposure (R-squared >0.9), while average wind speed/directions and some other meteorological factors are either non-significant or relatively insubstantial to predict annual average impacts. Since population sizes within different ranges from a power plant are the independent variables in the regression, changes in population from the

early 2000s to now are taken into account by multiplying the present population sizes and corresponding IF coefficients. In future studies, it is possible to estimate the IF coefficients with a larger sample to improve accuracy.”

Second, in the revised manuscript, we use GEOS-Chem model to statistically test the accuracy of the IF model. GEOS-Chem is a state-of-the-art three-dimensional air quality model that takes into account fine-scale emissions, atmospheric chemistry, aerosol microphysics, precipitation, wind speed and direction, and other input data (http://acmg.seas.harvard.edu/geos/geos_overview.html). We added a sub-section in main text Page 17:

“

Comparison of results based on different models

Intake fractions (IF) model is time-efficient to process more than 2000 simulations for all coal-fired plants in China, and the model parameters are calibrated specifically for China’s power plants, though it only considers variables that have the strongest predictive power. We use the GEOS-Chem model to statistically test the accuracy of the IF model. GEOS-Chem is a three-dimensional air quality model that takes into account fine-scale emissions, atmospheric chemistry, aerosol microphysics, precipitation, wind speed and direction, and other input data. Identical national power plants inventories are used in IF and GEOS-Chem simulations. The simulation results show that national total health losses due to coal-fired plants in 2017 was 460.1 billion USD based on IF model, and was 426.4 billion USD based on GEOS-Chem (Supplementary Figures 5 and 6). We then calculate health losses suffered by each province in China based on the two models (Figure 6). IF model results have higher values in remote regions, such as Heilongjiang, Inner Mongolia, Yunnan, and Tibet, while the GEOS-Chem model results have higher values in central provinces, such as Hubei, Hunan, and Sichuan. In majority of the other provinces, the values from the two models are close. We also compare GEOS-Chem simulation results with the actual observation data from 1444 air quality monitoring stations across China (Supplementary Figure 7). Supplementary Figure 7 indicates that GEOS-Chem tends to underestimate PM_{2.5} concentrations in remote provinces, while overestimate PM_{2.5} concentrations in central provinces. Thus, the large differences between the two models’ results in the seven provinces mentioned above can be partly explained by GEOS-Chem’s biases. Next, we calculate health losses suffered by each of the 2337 Chinese counties according to the two models, and run linear regressions for counties within each of the six regional power grids (Figure 7). All the six regressions show good correlation between the results based on the two models (R-squared range from 0.73 to

0.91). Thus, overall, we think that the IF model has the advantage of time efficiency in processing a large number of simulations and its results are reliable for policy analysis at large geographic scales. If the government wants to calculate an externality tax for an individual plant based on more comprehensive variables, then the IF model is not sufficiently customized, and a more sophisticated model, such as GEOS-Chem, should be used.

Figure 6. Values of provincial health losses in 2017 based on intake fractions (IF) model and GEOS-Chem model.

Figure 7. Linear regressions between results based on intake fractions model and GEOS-Chem model. a-f present the results in six regional power grids, respectively.

”

We also added discussion in Supplementary Note 4:

“We created a national inventory for pollutants emissions from all coal-fired power plants in China, and use GEOS-Chem and intake fractions (IF) models to simulate the overall impacts of these power plants in 2017, in terms of the increase in exposure to PM_{2.5} and the corresponding values of health losses. Supplementary Figures 5 and 6 present the distribution of health losses in 1*1km grids, based on IF and GEOS-Chem models, respectively. The simulation results show that national total health losses due to coal-fired plants in 2017 was 460.1 billion USD based on IF model, and was 426.4 billion USD based on GEOS-Chem; while the latter was 7.3% lower than the former, the maximum value from GEOS-Chem model results was higher than that of the IF model.

We also compared GEOS-Chem simulation results with the actual observation data from 1444 air quality monitoring stations across China (Supplementary Fig. 7). The result indicates that GEOS-Chem tends to underestimate PM_{2.5} concentrations in remote provinces (the blue dots), while overestimate PM_{2.5} concentrations in central provinces (red dots). More discussion of the model comparisons is presented in the Results Section in the main text.

Supplementary Fig. 5 Distribution of health losses due to air pollution caused by coal-fired power plants in 2017, based on intake fractions (IF) model. For each 1x1 km grid, the value represents the total health loss suffered by the population living in the grid, induced by air pollutants emitted from all coal-fired power plants in China in 2017. (Unit is in USD)

Supplementary Fig. 6 Distribution of economic losses due to air pollution caused by coal-fired power plants in 2017, based on GEOS-Chem model. For each 1x1 km grid, the value represents the total health loss suffered by the population living in the grid, induced by air pollutants emitted from all coal-fired power plants in China in 2017. (Unit is in USD)

Supplementary Fig. 7 Comparison of GEOS-Chem simulated annual PM_{2.5} concentration values with the actual observation data. Normalized mean bias is calculated as (simulated value-observed value)/observed value.

”

The valuation of damages is based on the US estimate of US\$7mil./person and scaled using an income elasticity of 0.5. This means that a population with 1/4 of the US income would have a VSL of \$7/2mil. And a person with 1/9 of the US income have a VSL of \$7/3million. The paper does not cite China-based VSL studies, I would say that none have such high valuations. Or more generally, studies of values of total China damages do not use such high numbers.

Response: We reviewed relevant literature and used different VSL values for comparison analyses. We find that similar studies, including Shindell 2012 ¹⁵, West 2013 ⁴, and Li 2018 ², used assumptions and calculation procedures that are close to our study, choosing the USEPA recommended VSL as baseline and adjusting VSLs using elasticities of 0.4 or 0.5. Cao et al. ²³ estimated China's VSL through a contingent valuation study and their recommended value is much lower relative to the aforementioned studies. But since the Cao et al. study is still in preprint status and has not been widely applied, we decide to use it in the SI as a comparison to our main results.

In main text, Page 31, Line 606, we added:

“As comparisons to the main results using a universal VSL based on national per capita GDP and an income elasticity of 0.5, we also calculated co-benefits using VSLs based on alternative assumptions, including 1) an income elasticity of 0.4; 2) prefecture-level VSLs based on local income; 3) VSL based on a contingent valuation survey study from China ²³. These results are presented in Supplementary notes 2&3.”

In the Supplementary Information, Page 5, we added:

“

Supplementary note 3:

The VSL used to calculate the results in the main text is based on: 1) China's national per capita GDP in 2017; 2) baseline VSL recommended by the United States Environmental Protection Agency (USEPA), which is 7.4 million USD per person in 2006 value; and 3) an income elasticity of 0.5.

We reviewed the VSL values used in similar studies, including: 1) Shindell et al. ¹⁵ employed the USEPA preferred VSL of \$9.5 million for 2030, and used an elasticity of 0.4 to estimate country-specific VSLs based on each country's income per capita; 2) Li et al. ² used the USEPA suggested VSL value and an elasticity of 0.4 to calculate

VSLs for each Chinese province based on the ratios of GDP per capita, similar to the procedure in Shindell et al.; 3) West et al.⁴ used both low VSLs and high VSLs: \$1.8 million for Western Europe (OECD recommended value) as the benchmark low estimate, and \$7.4 million for the U.S. (USEPA recommended value) as the benchmark high estimate, which are both adjusted to different world regions and into the future using an income elasticity of 0.5; 4) Cao et al.²³ estimated China's VSL through a contingent valuation study in six representative cities; they recommended a mean VSL value for China, which is 5.1 million CYN in 2019 price, or 725,000 USD in 2017 value.

Here we calculate co-benefits using alternative VSLs and elasticities: 1) China's national VSL based on the USEPA recommended benchmark VSL (7.4 million USD per person in 2006 value) and an income elasticity of 0.4, following Shindell et al.; 2) China's national VSL based on Cao et al.'s contingent valuation study, which is 5.1 million CYN in 2019 price, or 725,000 USD in 2017 value; 3) prefecture-level VSLs based on local income, using the USEPA recommended benchmark VSL and an elasticity of 0.5 (corresponding to Supplementary Fig. 2); 4) same assumptions as (3) but giving higher weights to local impacts (corresponding to Supplementary Fig. 3). These results are presented in Supplementary Figures 4a-4d, respectively. The results indicate that while different choices of VSLs have significant impact on co-benefit values, they do not significantly change the ranking of the values. Therefore, choosing different VSLs will not change the priority order for coal plants phasing out or retrofitting, but it will change the ratios between carbon costs and air pollution costs and affect the stringency of coal restriction policies.

Supplementary Fig. 4 Provincial co-benefits based on alternative VSLs and elasticities. (Unit is in USD) a. national VSL based on the USEPA recommended benchmark VSL and an elasticity of 0.4; b. national VSL using 725,000 USD in 2017 value; c. prefecture-level VSLs based on local income, using the USEPA recommended benchmark VSL and an elasticity of 0.5; d. same assumptions as c but giving higher weights to local impacts.

”

The paper commented that Beijing and surrounding areas do not have the highest estimated benefits of coal-use reductions but has the strictest control policies and suggest that it is sub-optimal. I think one should note that a prime motivator for the strict control in the 2000's was the run-up to the Beijing Olympics, and many would argue that it was indeed a proper national priority.

Response: We agree that environmental concerns are in many cases not the most important factor in policy decisions relative to socio-political considerations. Nevertheless, our calculation based on environmental benefits can serve as a benchmark for actual policy making. We added a caveat in the Discussion section to

acknowledge this point, on Page 23, Line 437:

“Socio-political considerations can outweigh climate and environmental concerns; for instance, as the national capital city, Beijing has the strictest coal control policies, even though it does not have the highest estimated co-benefits. Such social and political decisions are beyond the scope of our analysis, though our calculation based on environmental benefits can serve as a benchmark for actual policy making.”

The paper suggests (page 18) that it is feasible to integrate carbon pricing and pollution taxes into one unified pricing scheme. I do not think it is so easy, not least since the authors do not cite anyone else with an explicit proposal. As I note above, the carbon price should be compared with the cost of replacing a coal kWh with a non-fossil kWh. This is an additional complex calculation. Given that FGD and SCR technologies are effective we should probably focus on keeping explicit SO₂ and NO_x prices. The benefit of reducing a ton CO₂ should indeed include the co-benefit of reducing pollution.

Response: We understand that such a policy is very challenging. In the revised manuscript, we changed “it is feasible” to “our results suggest that it is possible”. Our purpose is just to point out that there is a possibility to unify the two types of policies, and China’s government agency re-organization can be an opportunity to explore such a unified policy specifically in the power sector. This is because since the major reorganization of central government bureaucracy in 2018, the Ministry of Ecology and Environment (MEE) has been in charge of both carbon emissions trading program and air pollution tax, and the power sector is the primary focus of both the policies. Of course we know there are many challenges in putting the policy in practice, so we discussed the real-world challenges in the following paragraph. In addition, we made a parallel between our proposal for trading ratios in the carbon emissions trading and Muller and Mendelsohn’s proposal to use trading ratios in SO₂ and NO_x emissions trading.

Our revised texts are presented on Page 21, starting from Line 391:

“Second, our study can help improve the design of China’s carbon pricing and pollution tax policies. In recent years China has been promoting market-based policies in climate and environmental fields ²⁴, including national and provincial carbon emissions trading systems and environmental pollution taxes. According to previous studies, the global social cost of one ton CO₂ emission is around US\$20-\$50 ²⁵. The social costs of air pollution are complicated by the heterogeneous influences of source locations and socioeconomic surroundings. The results of our study provide locational health co-

benefits and suggest that it is possible to integrate carbon pricing and pollution taxes into one unified pricing scheme, which could be either an emissions trading system or a tax. Muller and Mendelsohn argue that setting location-specific trading ratios in SO₂ and NO_x emissions trading systems can achieve significant economic benefits¹⁹. In parallel, our study suggests that similar trading ratios in a carbon emissions trading system can enhance the achievements in health co-benefits. For instance, to compensate for one ton CO₂ emission from a plant in Henan, the plant owner will need to buy allowances for two tons of CO₂ emissions from Ningxia, or three tons from Qinghai, since the co-benefit value is much higher in Henan. Similarly, under a unified emissions tax policy, one ton carbon emission in Henan would be taxed twice as much as one ton in Ningxia, or three times as much in Qinghai. Since the MEE is in charge of measuring and pricing both carbon and air pollutants emissions, and the carbon and air pollutants charges are imposed on the same power plants, a unified pricing scheme can streamline the administrative processes, and send market signals that can more comprehensively reflect the social costs of coal power.”

And the following paragraph on Page 22, starting from Line 415 discusses the four types of real-world challenges of our policy proposal:

“A series of challenges need to be addressed in order to better integrate our analysis into actual policies...”

Minor points.

1. The reference item 4 is listed as “China Electricity Association ...” Is this correct? Do you mean the China Electricity Council? If it is in Chinese it should be so stated, with the title in pinyin given too.

Response: We corrected the reference and added the pinyin title. Now the reference reads:

“8 China Electricity Council. *China's electric power industry annual development report (in Chinese, Zhongguo dianli hangye niandu fazhan baogao)*. (China Market Press, 2018).”

2. P6. “...from US\$45-236 ...” should have a “per ton CO2” to make that clear.

Response: We made the relevant change on Page 7, Line 138:

“...the building of a power plant at different locations will yield vastly different co-benefit values, ranging from US \$51-278 per ton CO₂ nationwide”.

3. P8. The provincial averages are computed using capacity weights. Since capacity utilization varies a bit, it would be more accurate to use output (MWh) as weights. Better still would be using fuel consumption as weights.

Response: We agree that it would be better to use fuel consumption or output as weights. However, we do not have facility-level fuel consumption or output data for all power plants. But since China’s coal power industry adopts the “equal shares dispatch” rule, meaning that all power plants in a province are assigned roughly the same operating hours each year, the capacity weights can also approximate each plant’s share in total electricity generation. We provided a reference to Kahrl et al. 2013 ²⁶ about the “equal shares dispatch” rule.

Relevant texts are presented on Page 33, Line 656:

“The provincial average co-benefit values were obtained by taking a capacity-weighted mean of co-benefit values for all individual plants in that province (Figure 2). Since China’s coal power industry adopts the “equal shares dispatch” rule, meaning that all power plants in a province are assigned roughly the same operating hours each year ²⁶, the weights can also approximate each plant’s share in total electricity generation.”

4. P19. In citing the Global Energy Monitor and the Ctr for Global Dev. as the sources of data, the full references of these sources should be given (book, website, etc). The source of meteorological data should be given.

Response: We added references of these sources as below:

29 Global Energy Monitor. *Global Coal Plant Tracker Project*, <<https://endcoal.org/global-coal-plant-tracker/>> (2021).

30 Center for Global Development. *Carbon Monitoring for Action (CARMA)*, <<https://www.cgdev.org/topics/carbon-monitoring-action>> (2012).

32 China Meteorological Data Service Centre. *Hourly data from surface meteorological stations in China*, <<http://data.cma.cn/en>> (2017).”

5. P22. “... Table 2, we calculated ΔC_{ijt} caused ...” I think you mean IFijr

Response: We made the relevant change on Page 26, Line 511:

“Using coefficients in Supplementary Table 2, we calculated $IF_{i,j,r}$ caused by SO_2 , NO_x , and primary $PM_{2.5}$ emissions based on equation (1).”

References

Fowlie, Meredith and N. Muller (2019) Market-Based Emissions Regulation When Damages Vary across Sources: What Are the Gains from Differentiation? Journal of the Association of Environmental and Resource Economists.

Muller, Nicholas and R. Mendelsohn (2009) Efficient Pollution Regulation: Getting the Prices Right, American Economic Review

Response: Thank you for recommending the two valuable references. We cited these two studies in multiple places in the manuscript to strengthen our analysis. They are #4 and #5 in the list of references.

References

- 1 Fowlie, M. & Muller, N. Market-based emissions regulation when damages vary across sources: What are the gains from differentiation? *Journal of the Association of Environmental and Resource Economists* **6**, 593-632 (2019).
- 2 Li, M. *et al.* Air quality co-benefits of carbon pricing in China. *Nature Climate Change* **8**, 398-403, doi:10.1038/s41558-018-0139-4 (2018).
- 3 Cai, W. *et al.* The Lancet Countdown on $PM_{2.5}$ pollution-related health impacts of China's projected carbon dioxide mitigation in the electric power generation sector under the Paris Agreement: a modelling study. *The Lancet Planetary Health* **2**, e151-e161 (2018).
- 4 West, J. J. *et al.* Co-benefits of Global Greenhouse Gas Mitigation for Future Air Quality and Human Health. *Nature Climate Change* **3**, 885-889, doi:10.1038/NCLIMATE2009 (2013).
- 5 IEA. CO2 Emissions from Fuel Combustion 2018, International Energy Agency. (2018).
- 6 Davidson, Michael R., Zhang, D., Xiong, W., Zhang, X. & Karplus, Valerie J. Modelling the potential for wind energy integration on China's coal-heavy electricity grid. *Nature Energy* **1**, doi:10.1038/nenergy.2016.86 (2016).
- 7 China Electricity Council. China's electric power industry annual development report (in Chinese, Zhongguo dianli hangye niandu fazhan baogao). (2019).
- 8 Tong, D. *et al.* Targeted emission reductions from global super-polluting power plant units. *Nature Sustainability* **1**, 59-68 (2018).
- 9 Karplus, V. J., Zhang, S. & Almond, D. Quantifying coal power plant responses to tighter SO_2 emissions standards in China. *Proceedings of the National Academy of Sciences* **115**, 7004-7009

- (2018).
- 10 Zheng, B. *et al.* Trends in China's anthropogenic emissions since 2010 as the consequence of
clean air actions. *Atmospheric Chemistry and Physics* **18**, 14095-14111 (2018).
- 11 Hammitt, J. K. & Robinson, L. A. The income elasticity of the value per statistical life:
transferring estimates between high and low income populations. *Journal of Benefit-Cost
Analysis* **2**, 1-29 (2011).
- 12 Liu, X. *et al.* Updated hourly emissions factors for Chinese power plants showing the impact of
widespread ultralow emissions technology deployment. *Environmental science & technology*
53, 2570-2578 (2019).
- 13 Zhang, H., Zhang, X. & Yuan, J. Coal power in China: A multi - level perspective review. *Wiley
Interdisciplinary Reviews: Energy and Environment*, e386 (2020).
- 14 Tang, L. *et al.* Substantial emission reductions from Chinese power plants after the introduction
of ultra-low emissions standards. *Nature Energy* **4**, 929-938 (2019).
- 15 Shindell, D. *et al.* Simultaneously mitigating near-term climate change and improving human
health and food security. *Science* **335**, 183-189 (2012).
- 16 Lelieveld, J., Evans, J. S., Fnais, M., Giannadaki, D. & Pozzer, A. The contribution of outdoor
air pollution sources to premature mortality on a global scale. *Nature* **525**, 367-371,
doi:10.1038/nature15371 (2015).
- 17 Hasselmann, K. *et al.* The challenge of long-term climate change. *Science* **302**, 1923-1925
(2003).
- 18 Parry, I., Veung, C. & Heine, D. How much carbon pricing is in countries' own interests? The
critical role of co-benefits. *Climate Change Economics* **6**, 1550019 (2015).
- 19 Muller, N. Z. & Mendelsohn, R. Efficient pollution regulation: getting the prices right. *American
Economic Review* **99**, 1714-1739 (2009).
- 20 International Energy Agency. World Energy Investment. (2020).
- 21 Zhao, C. *et al.* The economics of coal power generation in China. *Energy Policy* **105**, 1-9 (2017).
- 22 Zhou, Y., Levy, J. I., Evans, J. S. & Hammitt, J. K. The influence of geographic location on
population exposure to emissions from power plants throughout China. *Environ Int* **32**, 365-373,
doi:10.1016/j.envint.2005.08.028 (2006).
- 23 Cao, C. *et al.* Estimating the Value of Statistical Life in China: A Contingent Valuation Study in
Six Representative Cities. Preprint at [https://assets.researchsquare.com/files/rs-
199197/v1_stamped.pdf?c=1612551473](https://assets.researchsquare.com/files/rs-199197/v1_stamped.pdf?c=1612551473). (2021).
- 24 Wang, P., Liu, L., Tan, X. & Liu, Z. Key challenges for China's carbon emissions trading
program. *Wiley Interdisciplinary Reviews: Climate Change* **10**, e599 (2019).
- 25 Ackerman, F. & Stanton, E. Climate risks and carbon prices: Revising the social cost of carbon.
Economics: The Open-Access, Open-Assessment E-Journal **6**, 10 (2012).
- 26 Kahrl, F., Williams, J. H. & Hu, J. The political economy of electricity dispatch reform in China.
Energy Policy **53**, 361-369, doi:10.1016/j.enpol.2012.10.062 (2013).

Peer Review comments, additional round review –

Reviewer #1 (Remarks to the Author):

The changes are satisfactory.

Reviewer #2 (Remarks to the Author):

The authors made substantial changes to the manuscript, which now includes more clarifications about the study's assumptions and how to interpret the results. All my comments were convincingly implemented; only for the relationship between polluting and polluted regions shown in Fig.3, I see more potential in demonstrating the mismatch. However, overall, the relative relationship is now presented in a more comprehensible way than before.

Reviewer #3 (Remarks to the Author):

(see next page)

Report on NCOMMS-20-36232A. Revision of "Location-specific co-benefits of carbon emissions reduction from coal-fired power plants in China"

The methods of the paper are sound, and the results are useful; the main message that environmental policies should vary by location is important. I, however, have some problems with the approach to writing the paper. First is the lack of clarity of the policy being simulated and thus I was not sure what benefits in Figure 4 mean exactly on first reading. Related to this lack of clarity about policy is the authors' repeated use of the term co-benefits and at the same time discussing desulfurization policies, NO_x control policies, low sulfur coal policies (p 13-15).

The co-benefits of reducing one ton of coal use are clear; we have the direct benefit of reducing CO₂ and the co-benefits of lower SO₂, NO_x, etc. What are the co-benefits of a desulfurization policy? If there is an uncounted climate effect of reducing SO₂ it is probably negative, less sulfates probably mean higher temperatures.

I think the strength of the method is that it can calculate the total benefits of many types of policies by location. This includes the carbon reduction policies (carbon pricing, renewable standards, green certificates, etc), FGD, SCR requirements, and other pollution control policies. There is no reason to limit the paper, and the title, to "co-benefits of carbon reduction." But if the authors do wish to limit themselves to carbon reduction, then do not talk about desulfurization policies and confuse the discussion.

Reading pages 13-16 of the main results in Figure 4 leave me scratching my head. Until one gets to the Methods section much later that it seems that the authors simulated the reduction of x tons of coal which reduce CO₂ by some marginal y tons. This has the simultaneous effect of reducing SO₂ by s tons, etc. But instead of stating this clearly, p13 and p15 talks about FGDs, SCRs, bag filters. Why talk about them when you are not simulating the benefit by location of reducing 1 marginal ton of SO₂ or NO_x? All that was needed is to say that "at the level of control actually achieved in 2017, the reduction of 1 ton of coal use also reduces SO₂ by s tons, etc." Also, I do not think it is most helpful to express benefits as per kWh; the benefits should be related to the policy, if the policy is to cut coal use, then it seems more natural to express as benefits per ton of coal cut. If the point is that a switch of 1kWh from coal to renewables provide these benefits, then it is reasonable to express as benefit/kWh, but then the policy should be discussed in such terms, and a presentation of the cost of such a change to renewables.

In the Discussion section on p20, instead of saying "... feasible approach to quantifying the co-benefits that come from a power plant ..." why not just say "quantifying the total benefits of reducing various pollutant emissions in any location..."

Finally, I think one should distinguish between reducing the use of 1 ton of coal from reducing 1 ton of CO₂ emissions. Right now, these are the same things in almost all cases, however, in future when we have CO₂ sequestration then these will be different things. With

CCS there would probably still be SO₂ and NO_x emissions and one would need a completely new set of calculations.

The difficulties of transferring VSL estimates from one country to another has been discussed for a long time. For China studies, this issue is noted as far back as World Bank (1997) *Clear Water, Blue Skies* (annex by Lvovsky and Hughes). The income elasticities obtained from cross-section results within a country (the 0.5 or 0.4 values noted in this paper) are not appropriate for transferring across countries, they lead to untenable high values for poor countries. This problem is shown by the huge range in Sup Fig 4. I realize that the studies cited (Shindell et al. and West et al.) do use these low elasticities, but these are studies that try to cover the whole world. A China study should be more careful, one could use an elasticity of 1 that many papers have chosen, at least to show the impact.

Minor points.

P4. When citing estimates like “\$340 billion in health costs...,” they should be put in context, say relative to GDP or \$ per ton of coal equivalent saved or something.

P8 (and P31). The authors inclusion of nearby costs other than PM₂₅ is approximated by multiplying the loss by 3. There should be some reference to other studies or some justification for choosing 3.

P9. “Values in Beijing ...not highest ... indicating a suboptimal policy ...” This is not a good way to put it; first, as stated in the Conclusion “social-political considerations can outweigh ..” Second, more importantly, these are based on national average incomes. It might be perfectly rational for the Beijing, Tianjin governments to value their health damages more highly given their higher-than-average incomes. Not to mention that, perhaps a more localized dispersion modeling may give a higher estimate than the iF approximation.

Response to reviewers' comments

Location-specific co-benefits of carbon emissions reduction from coal-fired power plants in China

Pu Wang, Cheng-Kuan Lin, Yi Wang, Dachuan Liu, Dunjiang Song, Tong Wu

We have benefitted significantly from the editors and reviewers' suggestions, and have made substantial changes to the manuscript according to reviewers' comments. Our response is organized as follows: Each comment is repeated in blue colour, italic texts, followed by a description of the response and the corresponding changes made in the manuscript (in quotes). We also provide a revised manuscript with all changes tracked to facilitate the editors and reviewers' evaluation of our revision. The page and line numbers referred below are the numbers after all the tracked changes are hidden or accepted in the manuscript file.

Reviewer #1 (Remarks to the Author):

The changes are satisfactory.

Response: We sincerely thank the reviewer for the helpful comments.

Reviewer #2 (Remarks to the Author):

The authors made substantial changes to the manuscript, which now includes more clarifications about the study's assumptions and how to interpret the results. All my comments were convincingly implemented; only for the relationship between polluting and polluted regions shown in Fig.3, I see more potential in demonstrating the mismatch. However, overall, the relative relationship is now presented in a more comprehensible way than before.

Response: We sincerely thank the reviewer for the time and comments. We think that after we remade the Fig. 3 according to your suggestion in the last round, the current Fig. 3 is sufficient to present our results based on location-specific co-benefits and can support our policy interpretation satisfyingly. Though it is possible to use certain statistic model to demonstrate this mismatch from another perspective, it is not the

focus of our study. Since you agree that the relationship is now presented in a more comprehensible way, we will keep the figure in the current form.

Reviewer #3 (Remarks to the Author):

Report on NCOMMS-20-36232A. Revision of Location-specific co-benefits of carbon emissions reduction from coal-fired power plants in China ”

The methods of the paper are sound, and the results are useful; the main message that environmental policies should vary by location is important. I, however, have some problems with the approach to writing the paper. First is the lack of clarity of the policy being simulated and thus I was not sure what benefits in Figure 4 mean exactly on first reading. Related to this lack of clarity about policy is the authors ’ repeated use of the term co-benefits and at the same time discussing desulfurization policies, NOx control policies, low sulfur coal policies (p 13-15).

The co-benefits of reducing one ton of coal use are clear; we have the direct benefit of reducing CO₂ and the co-benefits of lower SO₂, NO_x, etc. What are the co-benefits of a desulfurization policy? If there is an uncounted climate effect of reducing SO₂ it is probably negative, less sulfates probably mean higher temperatures.

Response: Thank you for the very helpful comments. We agree that the simulations and policies around Figure 4 require more clarification, and that we should not extensively discuss the FGD, SCR, and FAB policies per se to confuse our discussion on co-benefits. We have made substantial revision to the section “Impacts of end-of-pipe technologies on co-benefits” to address the two points you raised. Please see our detailed response to the specific comments on Figure 4.

As to your point on the term “co-benefit”, in the revised manuscript, we give an explicit definition used in this study, which refers to the co-benefit of climate policy in air quality improvement and the related health gains. As you pointed out in a later comment, in China’s power sector, without the presence of carbon sequestration, CO₂ emission reduction is equivalent to reducing coal use in electricity generation that can simultaneously reduce emissions of SO₂, NO_x, and primary PM_{2.5}. It should be noted that we do not consider the co-benefits of desulfurization or denitrification policies in climate effect.

On Page 2, Line 43, we added:

“Reducing CO₂ emissions from electric power generation will simultaneously reduce the emissions of sulphur dioxide (SO₂), nitrogen oxides (NO_x), and primary PM_{2.5}. While the effects of carbon emissions are independent of source locations, the effects of air pollutant emissions are highly dependent on source locations, population distribution, climate, and other factors^{1,2}. Thus, spatially nuanced policies are key to

maximizing the co-benefit of climate policy in air quality improvement and the related health gains.”

I think the strength of the method is that it can calculate the total benefits of many types of policies by location. This includes the carbon reduction policies (carbon pricing, renewable standards, green certificates, etc), FGD, SCR requirements, and other pollution control policies. There is no reason to limit the paper, and the title, to “co-benefits of carbon reduction.” But if the authors do wish to limit themselves to carbon reduction, then do not talk about desulfurization policies and confuse the discussion.

Response: It is true that our methods can calculate the location-specific benefits of a series of policies aiming to reduce emissions of CO₂, SO₂, NO_x, and primary PM_{2.5}. There are two key reasons for our paper to focus on health co-benefits of carbon reduction. First, our research field is climate policy, and our main focus is on means to facilitate carbon emissions reduction. We believe that for developing countries that face severe air pollution threats, considering air quality co-benefits of carbon reduction will significantly enhance the motivation in climate mitigation, and a series of previous research (e.g. Shindell 2012 ³, West 2013 ⁴, Lelieveld 2015 ⁵, Li 2018 ⁶) has already established this concept. Air pollution per se is not the central topic of our study. Second, we would like to give an unified environmental indicator for the purpose of setting priority order for regions to phase out coal power. The co-benefit of per ton CO₂ reduction, in our opinion, is an appropriate indicator that integrates the costs of CO₂, SO₂, NO_x, and primary PM_{2.5} (after adding a carbon price). Having separate location-specific cost estimates for per ton SO₂, NO_x, and PM_{2.5} emissions do not serve this goal satisfyingly.

Hope you can agree that we analyse this issue from a different perspective based on our research field and objective, though it might not sound straightforward from an air pollution policy perspective.

Reading pages 13-16 of the main results in Figure 4 leave me scratching my head. Until one gets to the Methods section much later that it seems that the authors simulated the reduction of x tons of coal which reduce CO2 by some marginal y tons. This has the simultaneous effect of reducing SO2 by s tons, etc. But instead of stating this clearly, p13 and p15 talks about FGDs, SCRs, bag filters. Why talk about them when you are not simulating the benefit by location of reducing 1 marginal ton of SO2 or NOx? All that was needed is to say that “at the level of control actually achieved in 2017, the reduction of 1 ton of coal use also reduces SO2 by s tons, etc.” Also, I do not think it is most helpful to express benefits as per kWh; the benefits should be related to the policy, if the policy is to cut coal use, then it seems more natural to express as benefits per ton of coal cut. If the point is that a switch of 1kWh from coal

to renewables provide these benefits, then it is reasonable to express as benefit/kWh, but then the policy should be discussed in such terms, and a presentation of the cost of such a change to renewables.

Response: Thank you for your comments. Based on your suggestions, we rewrote the section “Impacts of end-of-pipe technologies on co-benefits”, and provided clarifications on the assumptions and the policies we simulated. Since the health co-benefit of per ton CO₂ emission reduction is calculated according to a power plant’s emission factors for CO₂, SO₂, NO_x, and primary PM_{2.5}, and the adoption of various end-of-pipe technologies can significantly change these emission factors, in this section we focus our analysis on the impacts of these technologies on co-benefits. We agree with you that we should not focus on the FGD, SCR, and FAB policies per se, so we revised the relevant texts.

In Figure 4a, we choose to express benefits as per kWh rather than per ton coal reduction for two reasons. First, coal quality can vary substantially across power plants and time. Reducing one ton coal in Chongqing is very different from reducing one ton coal in Shanghai, for instance, because of the vastly different ash and sulfur content. Our dataset that reflects this difference (based on the MEIC inventory developed by Tsinghua University) is in the format of emission factors, expressed as grams per kWh. Even though it is possible to convert the emission factors to emissions per ton coal reduction, it is an unnecessary step that has no effect on the final results. Second, due to the widespread deployment of pollutant removal technologies in China’s power sector, the abated emission factors are very different from the unabated emission factors. It is much easier to analyse the actual emissions rates in power sector in terms of emissions per kWh, rather than emissions per ton coal use.

We also added comparison of levelized cost of electricity (LCOE) for coal power and wind and solar power in China (US cents per kWh), and argue that adding a carbon price and air pollution costs to coal power price is important to help enhance the relative market competitiveness of wind and solar power and reduce coal power. Though we did not have extensive discussion on the LCOE, since it is not the key theme of our paper. On Page 20, starting from Line 384, we wrote:

“Coal-fired power plants are among the largest sources of CO₂ and air pollutants worldwide, but without policy interventions, they remain the most affordable and accessible electricity providers in many developing countries. The co-benefit value of per ton CO₂ reduction plus a carbon price can serve as a unified environmental indicator that enable policy makers to more accurately understand the social costs of electricity generation from coal burning, and to better address regional energy planning and environmental policymaking related to power plants. International Energy Agency (IEA)’s study estimated that in 2020, without a carbon price, the levelized cost of electricity (LCOE) for coal power in China was 5.2 US cents per

kWh, while the LCOE for wind and solar power were 5.8 and 5.1 US cents per kWh, respectively ⁷. Thus, adding a carbon price and air pollution costs to coal power price is important to help enhance the relative market competitiveness of wind and solar power and reduce coal power.”

The revised subsection “Impacts of end-of-pipe technologies on co-benefits” starts from Page 13, Line 234:

“The health co-benefit of per ton CO₂ emission reduction is calculated according to a power plant’s emission factors for CO₂, SO₂, NO_x, and primary PM_{2.5}, measured as grams per kWh electricity generation. These emission factors are determined by coal quality (e.g. ash and sulfur contents of coal), boiler types, and end-of-pipe pollutant removal technologies. In this section we analyse the impacts of four representative technologies that can significantly change these emission factors, including: (1) flue gas desulfurization (FGD) that can reduce up to 95% of SO₂ emissions, (2) selective catalytic reduction (SCR) that can reduce up to 85% of NO_x emissions, (3) switching from electrostatic precipitators to bag filters (or fabric filters, FAB) that increase the removal efficiency for particulate matters from 93% to 99%, and (4) switching from high-sulfur coal to coal with the national average sulfur content. In this analysis we do not consider any end-of-pipe carbon removal technology, because even though carbon capturing and storage (CCS) technology can substantially change CO₂ emission factors, it is not widely used in coal power sector in China by far. In the future when CO₂ sequestration is widely applied, we would need to take CCS into consideration in our calculations.

At the level of control actually achieved in 2017, the reduction of one kWh coal-fired electricity can reduce 309 grams of standard coal consumption on national average, which simultaneously reduces emissions of 840 g CO₂, 0.43 g SO₂, 0.98 g NO_x, and 0.15 g primary PM_{2.5}. Figure 4a presents the air-quality-related health benefit of reducing one kWh coal-fired electricity generation in each province, first based on unabated emission factors for SO₂, NO_x, and primary PM_{2.5}, then based on the ideal operation of FDG, SCR, ESP, and FAB that have removal efficiency of 95%, 85%, 93%, and 99%, respectively, and then based on 2017 actual emission factors. Regional-grid-level CO₂ emission factors in 2017 are given as well. Based on the ratio of carbon and air pollutants emitted per kWh electricity generation, we convert the health benefit values in Figure 4a to co-benefits per ton CO₂ emission reduction in Figure 4b.

Figures 4a and 4b show that the co-benefits patterns in Figures 1 and 2 are not fixed, and should be updated regularly (e.g. annual update) based on changes in penetration rates of different technologies in provinces. From a policy perspective, a province can deploy more end-of-pipe control technologies to change the carbon-air pollutants ratio, thus can change its relative priority in national coal plants phasing out; it can also receive a more favourable policy in a permit trading system or a tax system (see our

policy discussion in the Discussion section). There is no one-size-fits-all optimal strategy for all the provinces, and detailed local information is needed to improve policy effectiveness at the provincial level. Our calculations in Figure 4 can provide basis for such updates. First, desulfurization has the biggest impacts on co-benefit values across the country. In particular, in the central and southern provinces where coal has high percentage of sulfur, desulfurization could reduce up to 80% of pollution costs and substantially change the co-benefit estimates, as in the case of Chongqing. Second, in the northeastern and eastern provinces where coal has lower sulfur contents, desulfurization would only reduce roughly 35-55% of pollution costs. Consequently, in these areas the application of NO_x and dust removal technologies become important. Third, in provinces where pollution costs are high, such as in Chongqing and Guizhou, switching to coal with better quality can also have a significant impact on co-benefit values. Overall, we find the national emissions reduction due to end-of-pipe control technologies was around 70% on average in 2017, which is consistent with previous studies ⁸.”

In the Discussion section on p20, instead of saying “.. feasible approach to quantifying the co-benefits that come from a power plant ...” why not just say “quantifying the total benefits of reducing various pollutant emissions in any location ...”

Response: Thank you for your suggestion. We agree that it is indeed more accurate and comprehensible to change the sentence as you suggested. We revised this sentence as the following on Page 20, Line 382:

“First, our results provide a technically feasible approach to quantifying the total benefits of reducing various pollutant emissions from a power plant in any location.”

Finally, I think one should distinguish between reducing the use of 1 ton of coal from reducing 1 ton of CO₂ emissions. Right now, these are the same things in almost all cases, however, in future when we have CO₂ sequestration then these will be different things. With CCS there would probably still be SO₂ and NO_x emissions and one would need a completely new set of calculations.

Response: We added the following sentence to clarify on this point:

On Page 13, Line 245, we added:

“In this analysis we do not consider any end-of-pipe carbon removal technology, because even though carbon capturing and storage (CCS) technology can substantially change CO₂ emission factors, it is not widely used in coal power sector in China by

far. In the future when CO₂ sequestration is widely applied, we would need to take CCS into consideration in our calculations.”

The difficulties of transferring VSL estimates from one country to another has been discussed for a long time. For China studies, this issue is noted as far back as World Bank (1997) Clear Water, Blue Skies (annex by Lvovsky and Hughes). The income elasticities obtained from cross-section results within a country (the 0.5 or 0.4 values noted in this paper) are not appropriate for transferring across countries, they lead to untenable high values for poor countries. This problem is shown by the high range in Sup Fig 4. I realize that the studies cited (Shindell et al. and West et al.) do use these low elasticities, but these are studies that try to cover the whole world. A China study should be more careful, one could use an elasticity of 1 that many papers have chosen, at least to show the impact.

Response: Thank you for your suggestion. We added the calculations based on an elasticity of 1 in the supplementary information (SI Fig. 4a and SI Fig. 5), and compared the results with our results in the main text based on an elasticity of 0.5. We find that even though the values based on elasticity of 1 are much lower than those based on elasticity of 0.5, the general pattern and relative priority of provinces have not changed. The choice of elasticity value mostly influences the stringency of coal control policy, as we said in Supplementary note 3:

“The results indicate that while different choices of VSLs and elasticities have significant impact on co-benefit values, they do not significantly change the ranking of the values. Therefore, choosing different VSLs and elasticities will not change the priority order for coal plants phasing out or retrofitting, but it will change the ratios between carbon costs and air pollution costs and affect the stringency of coal restriction policies.”

And in the Discussion:

“...policy makers need to make various political and ethical decisions about valuation parameter choices and fairness concerns. Choosing different benchmark values of statistical life (VSLs) or elasticities will significantly change the values of co-benefits, thus affecting the stringency of coal restriction policies (Supplementary Figure 4); ... Such social and political decisions are beyond the scope of our analysis, though our calculation based on environmental benefits can serve as a benchmark for actual policy making.”

Below are the relevant changes made to the Supplementary Figures 4&5, based on an elasticity of 1:

Supplementary Fig. 4 Provincial co-benefits based on alternative VSLs and elasticities. (Unit is in USD) a. national VSL based on the USEPA recommended benchmark VSL and an elasticity of 1; b. national VSL using 725,000 USD in 2017 value; c. prefecture-level VSLs based on local income, using the USEPA recommended benchmark VSL and an elasticity of 0.5; d. same assumptions as c but giving higher weights to local impacts.

Supplementary Figure 5: Health co-benefits of per-ton CO₂ emissions reduction based on the USEPA recommended benchmark VSL and an elasticity of 1. (Unit is in USD).

Minor points.

P4. When citing estimates like “\$340 billion in health costs..,” they should be put in context, say relative to GDP or \$ per ton of coal equivalent saved or something.

Response: We have put such values into context. On Page 4, Line 81, we added:

“Li et al. applied global models to the latest Chinese data and estimated that a 4% annual reduction in energy intensity would lead to a 24% reduction in CO₂ emissions in 2030 relative to the no policy case; such a reduction would require a carbon price of \$72 per ton and lead to \$340 billion in avoided health costs in 2030.”

(Note that Li et al 2018 did not give the GDP projection in 2030, so we provided the carbon reduction and carbon price as the context.)

P8 (and P31). The authors inclusion of nearby costs other than PM25 is approximated by multiplying the loss by 3. There should be some reference to other studies or some justification for choosing 3.

Response: We added justification for the choice of 3, and further explained that policy makers can choose different weight values if they want to set stricter rules for

power plant construction in metropolitan regions:

On Page 31, Line 623:

“On national average, the distribution of economic loss caused by a plant across the four ranges (within 100km, 100-500km, 500-1000km, and beyond 1000km) is roughly 1:3:3:3. Since a power plant can have many detrimental effects on adjacent areas other than PM_{2.5} (e.g., coal ash and heavy metal deposition), the economic loss within 100km could be underestimated. As a comparison with the original results, in Supplementary Figure 3 we multiply the economic loss within 100km by 3 to make it roughly the same as the losses in other ranges. The choice of weight for local impacts can be adjusted based on various environmental and socioeconomic considerations, and from a regional planning perspective, policy makers can give higher weights to local impacts in order to avoid construction of coal plants in metropolitan areas.”

P9. “Values in Beijing ...not highest ... indicating a suboptimal policy ...” This is not a good way to put it; first, as stated in the Conclusion “social-political considerations can outweigh ..” Second, more importantly, these are based on national average incomes. It might be perfectly rational for the Beijing, Tianjin governments to value their health damages more highly given their higher-than-average incomes. Not to mention that, perhaps a more localized dispersion modeling may give a higher estimate than the iF approximation.

Response: We agree with you that the original argument “... indicating a suboptimal policy” can be misleading. So we decide to delete the text “indicating a suboptimal policy”, and revise the text as:

“Values in Beijing (\$180), Tianjin (\$183), and Hebei (\$183) are not among the highest, even though these provinces had the strictest policies for coal plants phasing-out or retrofitting; using VSLs based on local GDP per capita and higher weights for local effects will yield relatively higher values for these provinces (Supplementary Fig. 4c and 4d).”

References

- 1 Fowlie, M. & Muller, N. Market-based emissions regulation when damages vary across sources: What are the gains from differentiation? *Journal of the Association of Environmental and Resource Economists* **6**, 593-632 (2019).
- 2 Muller, N. Z. & Mendelsohn, R. Efficient pollution regulation: getting the prices

right. *American Economic Review* **99**, 1714-1739 (2009).

- 3 Shindell, D. *et al.* Simultaneously mitigating near-term climate change and improving human health and food security. *Science* **335**, 183-189 (2012).
- 4 West, J. J. *et al.* Co-benefits of Global Greenhouse Gas Mitigation for Future Air Quality and Human Health. *Nature Climate Change* **3**, 885-889, doi:10.1038/NCLIMATE2009 (2013).
- 5 Lelieveld, J., Evans, J. S., Fnais, M., Giannadaki, D. & Pozzer, A. The contribution of outdoor air pollution sources to premature mortality on a global scale. *Nature* **525**, 367-371, doi:10.1038/nature15371 (2015).
- 6 Li, M. *et al.* Air quality co-benefits of carbon pricing in China. *Nature Climate Change* **8**, 398-403, doi:10.1038/s41558-018-0139-4 (2018).
- 7 International Energy Agency. World Energy Investment. (2020).
- 8 Zheng, B. *et al.* Trends in China's anthropogenic emissions since 2010 as the consequence of clean air actions. *Atmospheric Chemistry and Physics* **18**, 14095-14111 (2018).

Final Reviewer round comments, reviewer 3:

Report on NCOMMS-275164_2. 2nd revision of "Location-specific co-benefits of carbon emissions reduction from coal-fired power plants in China"

This revision clarifies the discussion of the methods and aim of the paper. I think this is good. I just have a couple of observations for the authors to think about.

The paper, and the reply to reviewers, argue that a location-specific carbon policy is important and hence the paper supply such a calculation of location-specific benefits of reducing a ton of CO₂. The Discussion section of the paper contains a good discussion of "challenges need to be addressed" to implement such a policy. However, I think there is a more fundamental difficulty. It will be good to think about why there are no such location-specific carbon price proposals in the world, whereas there are proposals to price SO₂ and NO_x by location. I think the answer is clear, the practical problems of a location-specific price for SO₂ are already very severe – plant owners will argue about their tax and ask for a plant-specific air dispersion modeling exercise, etc.

A plant-specific carbon price will multiply these difficulties. Plant owners will be arguing over the modeling of SO₂, NO_x, PM, Hg, etc.; all at the same time. Every change of coal quality will bring a request for a new calculation. In China it won't be just companies arguing but local governments too. It seems much safer to argue that these location-specific price of CO₂ helps guide investment approvals, relocation support and shutdown priorities.

A small point about the reply regarding the use of benefits per kWh instead of benefits per ton of coal. I don't think the reason really matters. The authors are using the average of 309g/kWh, regional averages of pollution control rate, average regional coal quality. It says explicitly that 1 average kWh is 840 g of CO₂. This makes it equivalent to giving a benefit per average quality ton of coal used. Incidentally, the discussion in the Introduction is about benefit per ton of CO₂, which makes good sense. Simply giving just the benefit/ton-CO₂ in Figure 4 would thus be sufficient. There may be good reasons to report the equivalent average benefit/kWh, e.g., if the policy costs are discussed in terms of 1 kWh moved from coal to renewables.

Response to reviewers' comments, final review round

Location-specific co-benefits of carbon emissions reduction from coal-fired power plants in China

Pu Wang, Cheng-Kuan Lin, Yi Wang, Dachuan Liu, Dunjiang Song, Tong Wu

We have addressed the points raised in reviewer #3's comments, and have made relevant changes to the manuscript. Our response is organized as following: each comment is repeated in blue colour, italic texts, followed by a description of the response and the corresponding changes made in the manuscript (in quotes). We also provide a revised manuscript with all changes tracked. The page and line numbers referred below are the numbers after all the tracked changes are hidden or accepted in the manuscript file.

Reviewer #3 (Remarks to the Author):

Report on NCOMMS-275164_2. 2nd revision of Location-specific co-benefits of carbon emissions reduction from coal-fired power plants in China ”

This revision clarifies the discussion of the methods and aim of the paper. I think this is good. I just have a couple of observations for the authors to think about.

Response: We sincerely appreciate your valuable comments during the peer-review process, which have helped us substantially improve our analyses and argumentation. We think the two points you raised below are important, and have made relevant clarifications in the manuscript to address the two issues. Please see our detailed response under the two points below.

The paper, and the reply to reviewers, argue that a location-specific carbon policy is important and hence the paper supply such a calculation of location-specific benefits of reducing a ton of CO₂. The Discussion section of the paper contains a good discussion of “challenges need to be addressed” to implement such a policy. However, I think there is a more fundamental difficulty. It will be good to think about why there are no such location-specific carbon price proposals in the world, whereas there are proposals to price SO₂ and NO_x by location. I think the answer is clear, the practical problems of a location-specific price for SO₂ are already very severe –

plant owners will argue about their tax and ask for a plant-specific air dispersion modeling exercise, etc.

A plant-specific carbon price will multiply these difficulties. Plant owners will be arguing over the modeling of SO₂, NO_x, PM, Hg, etc.; all at the same time. Every change of coal quality will bring a request for a new calculation. In China it won't be just companies arguing but local governments too. It seems much safer to argue that these location-specific price of CO₂ helps guide investment approvals, relocation support and shutdown priorities.

Response: Thank you for the insightful comments. We agree that the application of plant-specific carbon price in real-world policies will still face practical challenges, due to uncertainties in modelling, frequent changes in coal quality, and the other issues we discussed in the “challenges need to be addressed” section. Therefore, as we stated in the Discussion section, our results demonstrate the possibility of precise policy making using plant-specific carbon price; in the future when better modelling techniques and real-time monitoring data are available, such a policy will become realistic. As you said, we also provide other two more realistic ways to apply the location-specific co-benefits: 1) in national energy planning, including the approval, retrofitting, and shutdown of coal-fired power plants in different locations; and 2) in provincial carbon pricing policies, such as emissions trading systems or carbon tax policies. Such differentiated provincial energy policies actually have precedents, such as the different provincial energy efficiency targets set by the central government in the twelfth and thirteenth Five-Year Plans.

We made relevant changes in the manuscript to better clarify our suggested applications of location-specific co-benefits. On Page 22, from Line 429:

“First, the current models still have large uncertainties in assessing exposure and health losses, which can undermine the benefits of location-specific policies. Even state-of-the-art models such as GEOS-Chem also show significant discrepancies between predicted and observed data. Therefore, our results demonstrate the possibility of precise policy making using plant-specific carbon price and co-benefits, though with relatively large uncertainties; in the future when better modelling techniques and real-time monitoring data are available, such a policy will become realistic. Currently, our results can be more realistically used in larger scale policy making, such as in national energy planning, including the approval, retrofitting, and shutdown of coal-fired power plants in different locations; and in provincially differentiated carbon pricing policies, such as emissions trading systems or carbon tax policies.”

A small point about the reply regarding the use of benefits per kWh instead of benefits per ton of coal. I don't think the reason really matters. The authors are using the average of 309g/kWh, regional averages of pollution control rate, average regional coal quality. It says explicitly that 1 average kWh is 840 g of CO₂. This makes it equivalent to giving a benefit per average quality ton of coal used. Incidentally, the discussion in the Introduction is about benefit per ton of CO₂, which makes good sense. Simply giving just the benefit/ton-CO₂ in Figure 4 would thus be sufficient. There may be good reasons to report the equivalent average benefit/kWh, e.g., if the policy costs are discussed in terms of 1 kWh moved from coal to renewables.

Response: We need to clarify that when we said “At the level of control actually achieved in 2017, the reduction of one kWh coal-fired electricity can reduce 309 grams of standard coal consumption on national average, which simultaneously reduces emissions of 840 g CO₂, 0.43 g SO₂, 0.98 g NO_x, and 0.15 g primary PM_{2.5}”, we were referring to the national average values to give readers a general sense of emissions from coal-fired power plants in China. Actually, in each province, the values vary due to differences in coal quality, energy efficiency of generators, and penetration rates of abatement technologies. Thus, even though the patterns in Figure 4a and 4b are similar, they are not exactly proportional. Providing the values based on emission factors of per kWh power generation first, and then converting the values to per ton CO₂, will make the connection between our data and results clearer, and make our co-benefit results more comprehensible for readers.

On Page 13, from Line 252, we revised the text as below:

“At the level of control actually achieved in 2017, the reduction of one kWh coal-fired electricity can reduce 309 grams of standard coal consumption on national average, which simultaneously reduces emissions of 840 g CO₂, 0.43 g SO₂, 0.98 g NO_x, and 0.15 g primary PM_{2.5}. These values vary in each province due to differences in coal quality, energy efficiency of generators, and penetration rates of abatement technologies. Figure 4a presents the air-quality-related health benefit of reducing one kWh coal-fired electricity generation in each province...Based on the ratio of carbon and air pollutants emitted per kWh electricity generation, we convert the health benefit values in Figure 4a to co-benefits per ton CO₂ emission reduction in Figure 4b.”